

# The Landslide Velocity

Shiva P. Pudasaini [a,b], Michael Krautblatter [a]

[a] Technical University of Munich, Chair of Landslide Research
Arcisstrasse 21, D-80333, Munich, Germany

[b] University of Bonn, Institute of Geosciences, Geophysics Section
Meckenheimer Allee 176, D-53115, Bonn, Germany
E-mail: shiva.pudasaini@tum.de

**Abstract:** Proper knowledge of velocity is required in accurately determining the enormous destructive energy carried by a landslide. We present the first, simple and physics-based general analytical landslide velocity model that simultaneously incorporates the internal deformation (non-linear advection) and externally applied forces, consisting of the net driving force and the viscous resistant. From the physical point of view, the model stands as a novel class of non-linear advective − dissipative system where classical Voellmy and inviscid Burgers' equation are specifications of this general model. We show that the non-linear advection and external forcing fundamentally regulate the state of motion and deformation, which substantially enhances our understanding of the velocity of a coherently deforming landslide. Since analytical solutions provide the fastest, the most cost-effective and the best rigorous answer to the problem, we construct several new and general exact analytical solutions. These solutions cover the wider spectrum of landslide velocity and directly reduce to the mass point motion. New solutions bridge the existing gap between the negligibly deforming and geometrically massively deforming landslides through their internal deformations. This provides a novel, rapid and consistent method for efficient coupling of different types of mass transports. The mechanism of landslide advection, stretching and approaching to the steady-state has been explained. We reveal the fact that shifting, up-lifting and stretching of the velocity field stem from the forcing and non-linear advection. The intrinsic mechanism of our solution describes the fascinating breaking wave and emergence of landslide folding. This happens collectively as the solution system simultaneously introduces downslope propagation of the domain, velocity up-lift and non-linear advection. We disclose the fact that the domain translation and stretching solely depends on the net driving force, and along with advection, the viscous drag fully controls the shock wave generation, wave breaking, folding, and also the velocity magnitude. This demonstrates that landslide dynamics are architectured by advection and reigned by the system forcing. The analytically obtained velocities are close to observed values in natural events. These solutions constitute a new foundation of landslide velocity in solving technical problems. This provides the practitioners with the key information in instantly and accurately estimating the impact force that is very important in delineating hazard zones and for the mitigation of landslide hazards.

## 1 Introduction

There are three methods to investigate and solve a scientific problem: laboratory or field data, numerical simulations of governing complex physical-mathematical model equations, or exact analytical solutions of simplified model equations. This is also the case for mass movements including extremely rapid flow-type landslide processes such as debris avalanches (Pudasaini and Hutter, 2007). The dynamics of a landslide are primarily controlled by the flow velocity. Estimation of the flow velocity is key for assessment of landslide hazards, design of protective structures, mitigation measures and landuse planning (Tai et al., 2001; Pudasaini and Hutter, 2007; Johannesson et al., 2009; Christen et al., 2010; Dowling and Santi, 2014; Cui et al., 2015; Faug, 2015; Kattel et al., 2018). Thus, a proper understanding of landslide velocity is a crucial requirement for an appropriate modelling of landslide impact force because the associated hazard is directly and strongly related to the landslide velocity (Huggel et al., 2005; Evans et al., 2009; Dietrich and Krautblatter, 2019). So, the landslide velocity is of great theoretical and practical interest for both scientists and engineers. However, the mechanical controls of the evolving velocity, runout and impact energy of the landslide have not yet been understood well.





Due to the complex terrain, infrequent occurrence, and very high time and cost demands of field measurements,
the available data on landslide dynamics are insufficient. Proper understanding and interpretation of the data
obtained from the field measurements are often challenging because of the very limited nature of the material
properties and the boundary conditions. Additionally, field data are often only available for single locations
and determined as static data after events. Dynamic data are rare (de Haas et al., 2020). So, much of the
low resolution measurements are locally or discretely based on points in time and space (Berger et al., 2011;
Schürch et al., 2011; McCoy et al., 2012; Theule et al., 2015; Dietrich and Krautblatter, 2019). Therefore,
laboratory or field experiments (Iverson et al., 2011; Iverson, 2012; de Haas and van Woerkom, 2016; Lu et
al., 2016; Lanzoni et al., 2017, Li et al., 2017; Pilvar et al., 2019; Baselt et al., 2021) and theoretical modelling
(Le and Pitman, 2009; Iverson and Ouyang, 2015; Pudasaini and Mergili, 2019) remain the major source of
knowledge in landslide and debris flow dynamics. Recently, there has been a rapid increase in the numerical
modelling for mass transports (McDougall and Hungr, 2005; Medina et al., 2008; Pudasaini, 2012; Cascini et
al., 2014; Cuomo et al., 2016; Frank et al., 2015; Iverson and Ouyang, 2015; Mergili et al., 2020a,b; Pudasaini
and Mergili, 2019; Qiao et al., 2019; Liu et al. 2021). However, to certain degree, numerical simulations are
approximations of the physical-mathematical model equations.
Although numerical simulations may overcome the limitations in the measurements and facilitate for a more
complete understanding by investigating much wider aspects of the flow parameters, run-out and deposition,
the usefulness of such simulations are often evaluated empirically (Mergili et al., 2020a, 2020b). In contrast,
exact, analytical solutions (Faug et al., 2010; Pudasaini, 2011) can provide better insights into the complex flow
behaviors, mainly the velocity, and their consequences. Moreover, analytical and exact solutions to non-linear
model equations are necessary to elevate the accuracy of numerical solution methods (Chalfen and Niemiec,
1986; Pudasaini, 2011, 2016; Pudasaini et al., 2018). For this reason, here, we are mainly concerned in pre-
senting exact analytical solutions for the newly developed general landslide velocity model equation.
Since Voellmy's pioneering work, several analytical models and their solutions have been presented in the liter-
ature for mass movements including extremely rapid flow-type landslide processes, avalanches and debris flows
(Voellmy, 1955; Salm, 1966; Perla et al., 1980; McClung, 1983). However, on the one hand, all these solutions
are effectively simplified to the mass point or center of mass motion. None of the existing analytical velocity
models consider advection or internal deformation. On the other hand, the parameters involved in these models
only represent restricted physics of the landslide material and motion. Nevertheless, a full analytical model that
includes a wide range of essential physics of the mass movements incorporating important process of internal
deformation and motion is still lacking. This is required for the more accurate description of landslide motion.
In the recent years, different analytical solutions have been presented for mass transports. These include sim-
ple and reduced analytical solutions for avalanches and debris flows (Pudasaini, 2011), two-phase flows (Ghosh
Hajra et al., 2017, 2018), landslide and avalanche mobility (Pudasaini and Miller, 2013; Parez and Aharonov,
2015), fluid flows in porous and debris materials (Pudasaini, 2016), flow depth profiles for mud flow (Di Cristo
et al., 2018), simulating the shape of a granular front down a rough incline (Saingier et al., 2016), the granular
monoclinal wave (Razis et al., 2018) and the mobility of submarine debris flows (Rui and Yin, 2019). However,
neither a more general landslide model as we have derived here, nor the solution for such a model exists in
literature.
This paper presents a novel non-linear advective - dissipative transport equation with quadratic source term as
a function of the state variable (the velocity) and their exact analytical solutions describing the landslide motion
down a slope. The source term represents the system forcing, containing the physical/mechanical parameters
and the landslide velocity. Our dynamical velocity equation largely extends the existing landslide models and
range of their validity. The new landslide velocity model and its analytical solutions are more general and
constitute the full description for velocities with wide range of applied forces and the internal deformation
associated with the spatial velocity gradient. In this form, and with respect to the underlying physics and
dynamics, the newly developed landslide velocity model covers both the classical Voellmy and inviscid Burgers
equation as special cases, but it also describes fundamentally novel and broad physical phenomena. Impor-
tantly, the new model unifies the Voellmy and inviscid Burgers' models and extends them further.





It is a challenge to construct exact analytical solutions even for the simplified problems in mass transport (Pudasaini, 2011, 2016; Di Cristo et al., 2018; Pudasaini et al., 2018). In its full form, this is also true for the landslide velocity model developed here. In contrast to the existing models, such as Voellmy-type and Burgers-type, the great complexity in solving the new model equation analytically derives from the simultaneous presence of the internal deformation (non-linear advection, inertia) and the quadratic source representing externally applied forces (in terms of velocity, including physical parameters). However, here, we advance further by constructing various analytical and exact solutions to the new general landslide velocity model by applying different advanced mathematical techniques, including those presented by Nadjafikhah (2009) and Montecinos (2015). We revealed several major novel dynamical aspects associated with the general landslide velocity model and its solutions. We show that a number of important physical phenomena are captured by the new solutions. Some special features of the new solutions are discussed in detail. This includes - landslide propagation and stretching; wave generation and breaking; and landslide folding. We also observed that different methods consistently produce similar analytical solutions. This highlights the intrinsic characteristics of the landslide motion described by our new model. As exact, analytical solutions disclose many new and essential physics, the solutions derived in this paper may find applications in environmental, engineering and industrial mass transport down slopes and channels.

## 2 Basic Balance Equation for Landslide Motion

### 2.1 Mass and momentum balance equations

A geometrically two-dimensional motion down a slope is considered. Let $t$ be time, $(x, z)$ be the coordinates and $(g^x, g^z)$ the gravity accelerations along and perpendicular to the slope, respectively. Let, $h$ and $u$ be the flow depth and the mean flow velocity along the slope. Similarly, $\gamma, \alpha_s, \mu$ be the density ratio between the fluid and the particles ($\gamma = \rho_f/\rho_s$), volume fraction of the solid particles (coarse and fine solid particles), and the basal friction coefficient ($\mu = \tan\delta$), where $\delta$ is the basal friction angle, in the mixture material. Furthermore, $K$ is the earth pressure coefficient as a function of internal and the basal friction angles, and $C_{DV}$ is the viscous drag coefficient.

We start with the multi-phase mass flow model (Pudasaini and Mergili, 2019) and include the viscous drag (Pudasaini and Fischer, 2020). For simplicity, we first assume that the relative velocity between coarse and fine solid particles ($u_s, u_{fs}$) and the fluid phase ($u_f$) in the landslide (debris) material is negligible, that is, $u_s \approx u_{fs} \approx u_f =: u$, and so is the viscous deformation of the fluid. This means, for simplicity, we are considering an effectively single-phase mixture flow. Then, by summing up the mass and momentum balance equations, we obtain a single mass and momentum balance equation describing the motion of a landslide as:

$$\frac{\partial h}{\partial t} + \frac{\partial}{\partial x}(hu) = 0, \tag{1}$$

$$\frac{\partial}{\partial t}(hu) + \frac{\partial}{\partial x}\left[hu^2 + (1-\gamma)\alpha_s g^z K \frac{h^2}{2}\right] = h\left[g^x - (1-\gamma)\alpha_s g^z \mu - g^z\{1-(1-\gamma)\alpha_s\}\frac{\partial h}{\partial x} - C_{DV}u^2\right], \tag{2}$$

where $-(1-\alpha_s)g^z\partial h/\partial x$ emerges from the hydraulic pressure gradient associated with possible interstitial fluids in the landslide. Moreover, the term containing $K$ on the left hand side and the other terms on the right hand side in the momentum equation (2) represent all the involved forces. The first term in the square bracket on the left hand side of (2) describes the advection, while the second term (in the square bracket) describes the extent of the local deformation that stems from the hydraulic pressure gradient of the free-surface of the landslide. The first, second, third and fourth terms on the right hand side of (2) are the gravity acceleration; effective Coulomb friction that includes lubrication ($1-\gamma$), liquefaction ($\alpha_s$) (because, if there is no or substantially low amount of solid, the mass is fully liquefied, e.g., lahar flows); the local deformation due to the pressure gradient; and the viscous drag, respectively. Note that the term with $1-\gamma$ or $\gamma$ originates from the buoyancy effect. By setting $\gamma = 0$ and $\alpha_s = 1$, we obtain a dry landslide, grain flow or an avalanche motion. For this choice, the third term on the right hand side vanishes. However, we keep $\gamma$ and $\alpha_s$ also to include possible fluid effects in the landslide (mixture).





### 2.2 The landslide velocity equation

The momentum balance equation (2) can be re-written as:

$$h \left[ \frac{\partial u}{\partial t} + u \frac{\partial u}{\partial x} \right] + u \left[ \frac{\partial h}{\partial t} + \frac{\partial}{\partial x} (hu) \right]$$
$$= h \left[ g^x - (1 - \gamma) \alpha_s g^z \mu - g^z \left\{ ((1 - \gamma) K + \gamma) \alpha_s + (1 - \alpha_s) \right\} \frac{\partial h}{\partial x} - C_{DV} u^2 \right]. \tag{3}$$

Note that for $K = 1$ (which mostly prevails for extensional flows, Pudasaini and Hutter, 2007), the third term on the right hand side associated with $\partial h / \partial x$ simplifies drastically, because $\{((1 - \gamma) K + \gamma) \alpha_s + (1 - \alpha_s)\}$ becomes unity. So, the isotropic assumption (i.e., $K = 1$) loses some important information about the solid content and the buoyancy effect in the mixture. Employing the mass balance equation (1), the momentum balance equation (3) can be re-written as:

$$\frac{\partial u}{\partial t} + u \frac{\partial u}{\partial x} = g^x - (1 - \gamma) \alpha_s g^z \mu - g^z \left\{ ((1 - \gamma) K + \gamma) \alpha_s + (1 - \alpha_s) \right\} \frac{\partial h}{\partial x} - C_{DV} u^2. \tag{4}$$

The gradient $\partial h / \partial x$ might be approximated, say as $h_g$, and still include its effect as a parameter that may be estimated. Here, we are mainly interested in developing a simple but more general landslide velocity model than the existing ones that can be solved analytically and highlight its essence to enhance our understanding of the landslide dynamics.

Now, with the notation $\alpha := g^x - (1 - \gamma) \alpha_s g^z \mu - g^z \left\{ ((1 - \gamma) K + \gamma) \alpha_s + (1 - \alpha_s) \right\} h_g$, which includes the forces: gravity; friction, lubrication and liquefaction; and surface gradient; and $\beta := C_{DV}$, which is the viscous drag coefficient, (4) becomes a simple model equation:

$$\frac{\partial u}{\partial t} + u \frac{\partial u}{\partial x} = \alpha - \beta u^2, \tag{5}$$

where $\alpha$ and $\beta$ constitute the net driving and the resisting forces in the system. We call (5) the landslide velocity equation.

### 2.3 A novel physical−mathematical system

Equation (5) constitutes a genuinely novel class of non-linear advective - dissipative system and involves dynamic interactions between the non-linear advective (or, inertial) term $u \partial u / \partial x$ and the external forcing (source) term $\alpha - \beta u^2$. However, in contrast to the viscous Burgers' equation where the dissipation is associated with the (viscous) diffusion, here, dissipation stems because of the viscous drag, $-\beta u^2$. In the form, (5) is similar to the classical shallow water equation. However, from the mechanics and the material composition, it is much wider as such model does not exist in the literature. From the physical and mathematical point of view, there are two crucial novel aspects associated with model (5). First, it explains the dynamics of deforming landslide and thus extends the classical Voellmy model (Voellmy, 1955; Salm, 1966; McClung, 1983; Pudasaini and Hutter, 2007) due to the broad physics carried by the model parameters, $\alpha, \beta$; and the dynamics described by the new term $u \partial u / \partial x$. These parameters and the term $u \partial u / \partial x$ control the landslide deformation and motion. Second, it extends the classical non-linear inviscid Burgers' equation by including the non-linear source term, $\alpha - \beta u^2$, as a quadratic function of the unknown field variable, $u$, taking into account many different forces associated with the system as explained in Section 2.2.

From the structure, (5) is a fundamental non-linear partial differential equation, or a non-linear transport equation with a source, where the source is the external physical forcing. Such an equation explains the non-linear advection with source term that contains the physics of the underlying problem through the parameters $\alpha$ and $\beta$. The form of this equation is very important as it may describe the dynamical state of many extended (as compared to the Voellmy and Burgers models) physical and engineering problems appearing in nature, science and technology, including viscous/fluid flow, traffic flow, shock theory, gas dynamics, landslide and avalanches (Burgers, 1948; Hopf, 1950; Cole, 1951; Nadjafikhah, 2009; Pudasaini, 2011; Montecinos, 2015).





## 3  The Landslide Velocity: Simple Solutions

Exact analytical solutions to simplified cases of non-linear debris avalanche model equations are necessary to calibrate numerical simulations of flow depth and velocity profiles. These problem-specific solutions provide important insight into the full behavior of the system. Physically meaningful exact solutions explain the true and entire nature of the problem associated with the model equation, and thus, are superior over numerical simulations (Pudasaini, 2011; Faug, 2015).

One of the main purposes of this contribution is to obtain exact analytical velocities for the landslide model (5). In the form (5) is simple. So, one may tempt to solve it analytically to explicitly obtain the landslide velocity. However, it poses a great mathematical challenge to derive explicit analytical solutions for the landslide velocity, $u$. This is mainly due to the new terms appearing in (5). Below, we construct five different exact analytical solutions to the model (5) in explicit form. In order to gain some physical insights into the landslide motion, the solutions are compared to each other. Equation (5) can be considered in two different ways: steady-state and transient motions, and both without and with (internal) deformation that is described by the term $u\partial u/\partial x$.

### 3.1  Steady−state motion

For a sufficiently long time and sufficiently long slope, the time independent steady-state motion can be developed. Then, (5) reduces to a simplified equation for the landslide velocity down the entire slope:

$$\frac{\partial}{\partial x}\left(\frac{1}{2}u^2\right) = \alpha - \beta u^2. \tag{6}$$

Equivalently, this also represents a mass point velocity along the slope. Classically, (6) is called the center of mass velocity of a dry avalanche of flow type (Perla et al., 1980).

#### 3.1.1  Negligible viscous drag

In situations when the Coulomb friction is dominant and the motion is slow, the viscous drag contribution can be neglected ($\beta u^2 \approx 0$), e.g., typically the moment after the mass release. Then, the solution to (6) is given by (**Solution A**):

$$u(x;\alpha) = \sqrt{2\alpha\left(x - x_0\right) + u_0^2}, \tag{7}$$

where $x$ is the downslope travel distance, and $u_0$ is the initial velocity at $x_0$ (or, a boundary condition). Solution (7) recovers the landslide velocity obtained by considering the simple energy balance for a mass point in which only the gravity and simple dry Coulomb frictional forces are considered (Scheidegger, 1973), both of these forces are included in $\alpha$. Furthermore, when the slope angle is sufficiently high or close to vertical, (7) also represents a near free fall landslide or rockfall velocity for which $x$ changes to the vertical height drop.

#### 3.1.2  Viscous drag included

In general, depending on the magnitude of the net driving force (that also includes the Coulomb friction), the viscous drag and the magnitude of the velocity, either $\alpha$ or $\beta u^2$, or both can play dominant role in determining the landslide motion. Then, the more general solution for (6) than (7) takes the form (**Solution B**):

$$u(x;\alpha,\beta) = \sqrt{\frac{\alpha}{\beta}\left[1 - \left(1 - \frac{\beta}{\alpha}u_0^2\right)\frac{1}{\exp(2\beta(x - x_0))}\right]}, \tag{8}$$

where, $u_0$ is the initial velocity at $x_0$. The velocity given by (8) can be compared to the Voellmy velocity and be used to calculate the speed of an avalanche (Voellmy, 1955; McClung, 1983). However, the Voellmy model only considers the reduced physical aspects in which $\alpha$ merely includes the gravitational force due to the slope and the dry Coulomb frictional force. This has been discussed in more detail in Section 3.2. As in (7), the solution (8) can also represent a near free fall landslide (or rockfall) velocity when the slope angle is sufficiently high or close to vertical, but now, it also includes the influence of drag, akin to the sky-jump.



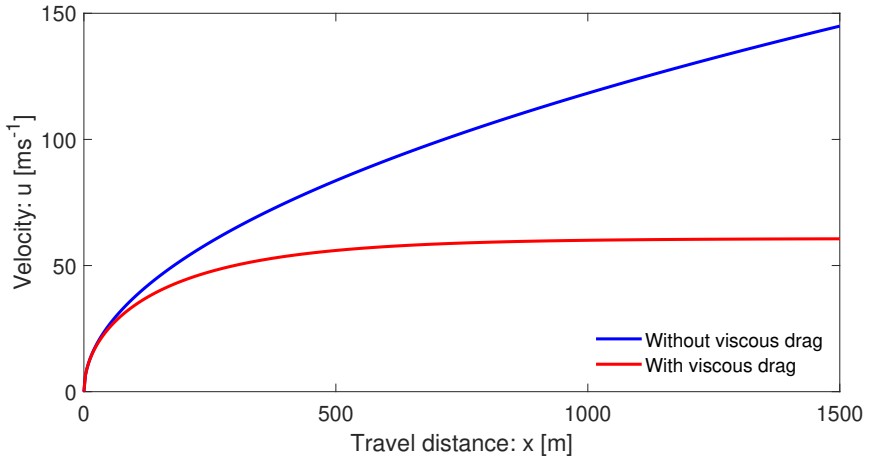

Figure 1: The landslide velocity distributions down the slope as a function of position, for both without and with drag given by (7) and (8), respectively. With drag, the flow attains the terminal velocity $u_{T^x} \approx 60.1$ ms$^{-1}$ at about $x = 600$ m, but without drag, the flow velocity increases unboundedly.

It is important to reveal the dynamics of viscous drag in the landslide motion. The major aspect of viscous drag is to bring the velocity (motion) to a terminal velocity (steady, uniform) for a sufficiently long travel distance. This is achieved by the following relation obtained from (8):

$$\lim_{x \to \infty} u = \sqrt{\frac{\alpha}{\beta}} =: u_{T^x}, \tag{9}$$

where $u_{T^x}$ stands for the terminal velocity of a deformable mass, or a mass point motion (Voellmy), along the slope that is often used to calculate the maximum velocity of an avalanche (Voellmy, 1955; McClung, 1983; Pudasaini and Hutter, 2007).

In what follows, unless otherwise stated, we use the plausibly chosen physical parameters for rapid mass movements: slope angle of about 50°, $\gamma = 1100/2700, \alpha_s = 0.65, \delta = 20°$ (Mergili et al., 2020a, 2020b; Pudasaini and Fischer, 2020). This implies the model parameters $\alpha = 7.0$, $\beta = 0.0019$. In reality, based on the physics of the material and the flow, the numerical values of these model parameters should be set appropriately. However, in principle, all of the results presented here are valid for any choice of the parameter set $\{\alpha, \beta\}$. For simplicity, $u_0 = 0$ is set at $x_0 = 0$, which corresponds to initially zero velocity at the position of the mass release. Figure 1 displays the velocity distributions of a landslide down the slope as a function of the slope position $x$. The magnitudes of the solutions presented here are mainly for reference purpose, which, however, are subject to scrutiny with laboratory or field data as well as natural events. For the order of magnitudes of velocities of natural events, we refer to Section 3.2.2. The velocities in Fig. 1 with and without drag, equations (7) and (8), respectively, behave completely differently already after the mass has moved a certain distance. The difference increases rapidly as the mass slides further down the slope. With the drag, the terminal velocity ($u_{T^x} = \sqrt{\alpha/\beta} \approx 60.1$ ms$^{-1}$) is attained at a sufficient distance (about $x = 600$ m). But, without drag, the velocity increases forever, which is less likely for a mass propagating down a long distance.

We note that as $\beta \to 0$, the solution (8) approaches (7). For relatively small travel distance, say $x \leq 50$ m, these two solutions are quite similar as the viscous drag is not sufficiently effective yet. However, for a long travel distance, $x \gg 50$ m, when the viscous drag in not included, the landslide velocity increases steadily without any control, whilst it increases only slowly, and remains almost unchanged for $x \geq 500$ m when the viscous drag effect is involved.





## 3.2 A mass point motion

Assume no or negligible local deformation (e.g., $\partial u/\partial x \approx 0$), or a Lagrangian description. Both are equivalent to the mass point motion. In this situation, only the ordinary differentiation with respect to time is involved, and $\partial u/\partial t$ can be replaced by $du/dt$. Then, the model (5) reduces to

$$\frac{du}{dt} = \alpha - \beta u^2. \tag{10}$$

Perla et al. (1980) also called (10) the governing equation for the center of mass velocity, however, for a dry avalanche of flow type. This is a simple non-linear first order ordinary differential equation. This equation can be solved to obtain exact analytical solution for velocity of the landslide motion in terms of a tangent hyperbolic function (**Solution C**):

$$u(t; \alpha, \beta) = \sqrt{\frac{\alpha}{\beta}} \tanh \left[ \sqrt{\alpha\beta}\,(t - t_0) + \tanh^{-1}\left( \sqrt{\frac{\beta}{\alpha}}\, u_0 \right) \right], \tag{11}$$

where, $u_0 = u(t_0)$ is the initial velocity at time $t = t_0$. Equation (11) provides the time evolution of the velocity of the coherent (without fragmentation and deformation) sliding mass until the time it fragments and/or moves like an avalanche. This transition time is denoted by $t_A$ (or, $t_F$) indicating fragmentation, or the inception of the avalanche motion due to fragmentation or large deformation. So, (11) is valid for $t < t_A$. For $t > t_A$, we must use the full dynamical mass flow model (Pudasaini, 2012; Pudasaini and Mergili, 2019), or the equations (1) and (2). For more detail on it, see Section 6.1.

For sufficiently long time, or in the limit as the viscous force brings the motion to a non-accelerating state (steady, uniform), from (11) we obtain:

$$\lim_{t \to \infty} u = \sqrt{\frac{\alpha}{\beta}} =: u_{T^t}, \tag{12}$$

where $u_{T^t}$ stands for the terminal velocity of the motion of a point mass.

**The landslide position:** Since $u(t) = dx/dt$, (11) can be integrated to obtain the landslide position as a function of time:

$$x(t; \alpha, \beta) = x_0 + \frac{1}{\beta} \ln \left[ \cosh \left\{ \sqrt{\alpha\beta}\,(t - t_0) - \tanh^{-1}\left( \sqrt{\frac{\beta}{\alpha}} u_0 \right) \right\} \right] - \frac{1}{\beta} \ln \left[ \cosh \left\{ -\tanh^{-1}\left( \sqrt{\frac{\beta}{\alpha}} u_0 \right) \right\} \right], \tag{13}$$

where $x_0$ corresponds to the position at the initial time $t_0$.

Figure 2 displays the velocity profile of a landslide down the slope as a function of the time as given by (11). For simplicity, $u_0 = 0$ is set as initial condition at $t_0 = 0$, which corresponds to initially zero velocity at the time of the landslide trigger. The terminal velocity $\left( u_{T^t} = \sqrt{\alpha/\beta} \right)$ is attained at a sufficiently long time ($\sim 15$ s). We note that, in the structure, the model (10) and its solution (11) exists in literature (Pudasaini and Hutter, 2007) and is classically called Voellmy's mass point model (Voellmy, 1955), or Voellmy-Salm model (Salm, 1966) that disregards the position dependency of the landslide velocity (Gruber, 1989). But, $(1 - \gamma)$, $\alpha_s$, and the term associated with $h_g$ are new contributions and were not included in the Voellmy model, and $K = 1$ therein, while in our consideration $\alpha$, $K$ can be chosen appropriately. Thus, the Voellmy model corresponds to the substantially reduced form of $\alpha$, with $\alpha = g^x - g^z \mu$.

### 3.2.1 The dynamics controlled by the physical and mechanical parameters

Solutions (8) and (11) are constructed independently, one for the velocity of a deformable mass as a function of travel distance, or the velocity of the center of mass of the landslide down the slope, and the other for the velocity of a mass point motion as a function of time. Unquestionable, they have their own dynamics. However,





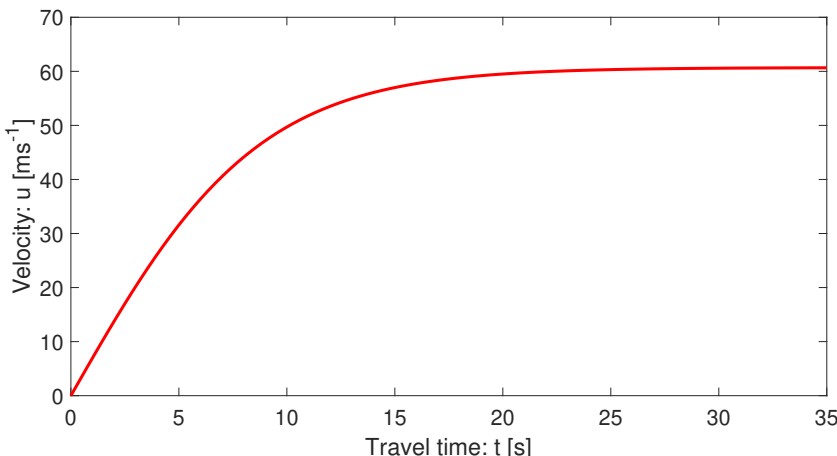

Figure 2: Time evolution of the landslide velocity down the slope with drag given by (11). The motion attains the terminal velocity at about $t = 15$ s.

for sufficiently long distance and sufficiently long time, or in the space and time limits, these solutions coincide
and we obtain a unique relationship:

$$u_{T^x} = u_{T^t} = \sqrt{\frac{\alpha}{\beta}}. \tag{14}$$

So, after a sufficiently long distance or a sufficiently long time, the forces associated with $\alpha$ and $\beta$ always
maintain a balance resulting in the terminal velocity of the system, $\sqrt{\alpha/\beta}$. This is a fantastic situation.
Intuitively this is clear because, one could simply imagine that sufficiently long distance could somehow be
perceived as sufficiently long time, and for these limiting (but fundamentally different) situations, there exists
a single representative velocity that characterizes the dynamics. This has exactly happened, and is an advanced
understanding. This has been shown in Fig. 3 which implicitly indicates the equivalence between (8) and (11).
In fact, this can be proven, because, for the mass point or the center of mass motion,

$$\frac{du}{dt} = \frac{du}{dx}\frac{dx}{dt} = u\frac{du}{dx} = \frac{du}{dx}\left(\frac{1}{2}u^2\right) = \frac{\partial u}{\partial x}\left(\frac{1}{2}u^2\right), \tag{15}$$

is satisfied.
In Fig. 3, both velocities have the same limiting values, but their early behaviours are quite different. In space,
the velocity shows hyper increase after the incipient motion. However, the time evolution of velocity is slow
(almost linear) at first, then fast, and finally attains the steady-state, $\sqrt{\alpha/\beta} = 60.1$ ms$^{-1}$, the common value
for both the solutions.
**3.2.2  The velocity magnitudes**
Importantly, for a uniformly inclined slope, the landslide reaches its maximum or the terminal velocity after a
relatively short travel distance, or time with value on the order of 50 ms$^{-1}$. These are often observed scenarios,
e.g., for snow or rock-ice avalanches (Schaerer, 1975; Gubler, 1989; Christen et al., 2002; Havens et al., 2014).
The velocity magnitudes presented above are quite reasonable for fast to rapid landslides and debris avalanches
and correspond to several natural events (Highland and Bobrowsky, 2008). The front of the 2017 Piz-Chengalo
Bondo landslide (Switzerland) moved with more than 25 ms$^{-1}$ already after 20 s of the rock avalanche release
(Mergili et al., 2020b), and later it moved at about 50 ms$^{-1}$ (Walter et al., 2020). The 1970 rock-ice avalanche
event in Nevado Huascaran (Peru) reached mean velocity of 50 - 85 ms$^{-1}$ at about 20 s, but the maximum
velocity in the initial stage of the movement reached as high as 125 ms$^{-1}$ (Erismann and Abele, 2001; Evans et



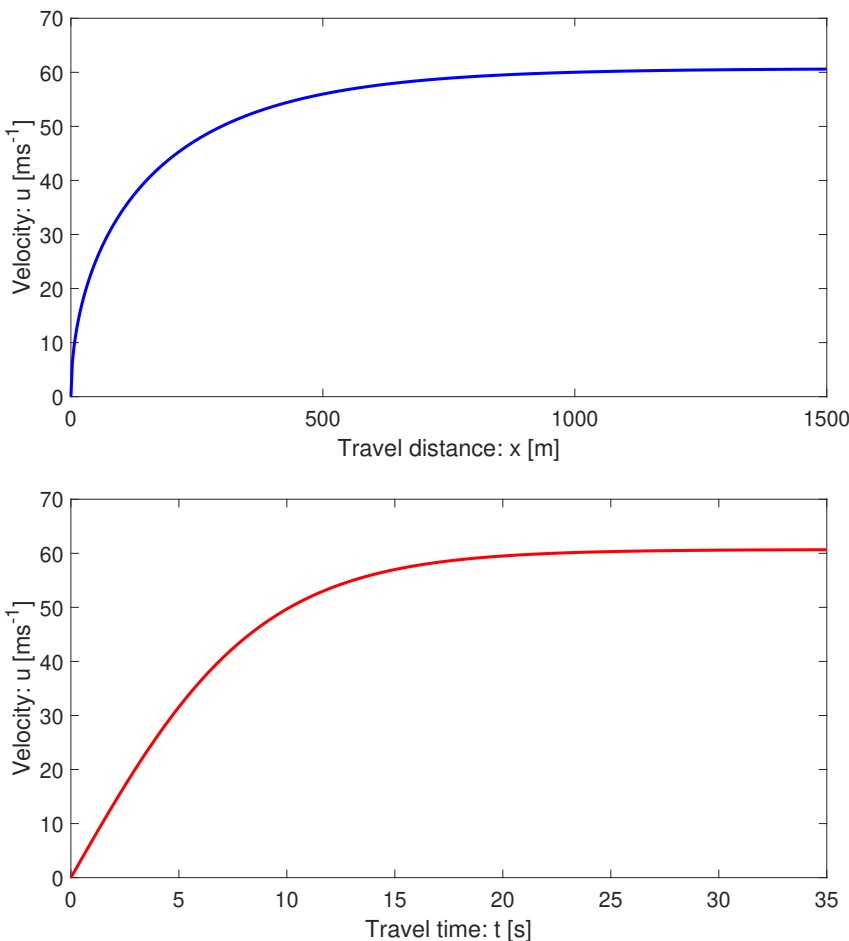

Figure 3: Evolution of the landslide velocity down the slope as a function of space (top) given by (8), and time (bottom) given by (11), respectively, both with drag. The flow attains the terminal velocity at about $x = 600$ m and $t = 15$ s.

al., 2009; Mergili et al. 2018). The 2002 Kolka glacier rock-ice avalanche in the Russian Kaucasus accelerated with the velocity of about 60 - 80 ms$^{-1}$, but also attained the velocity as high as 100 ms$^{-1}$, mainly after the incipient motion (Huggel et al., 2005; Evans et al., 2009).

### 3.2.3 Accelerating and decelerating motions

Depending on the magnitudes of the involved forces, and whether the initial mass was released or triggered with a small (including zero) velocity or with high velocity, e.g., by a strong seismic shacking, (11) provides fundamentally different but physically meaningful velocity profiles. Both solutions asymptotically approach $\sqrt{\alpha/\beta}$, the lead magnitude in (11). For notational convenience, we write $S_n(\alpha, \beta) = \sqrt{\alpha/\beta}$, which has the dimension of velocity, $\sqrt{\alpha/\beta}$ and is called the separation number (velocity) as it separates accelerating and decelerating regimes. Description for deceleration is given below. Furthermore, $S_n$ includes all the involved forces in the system and is the function of the ratio between the mechanically known forces: gravity, friction, lubrication and surface gradient; and the viscous drag force. Thus, $S_n$ fully governs the ultimate state of the landslide motion.





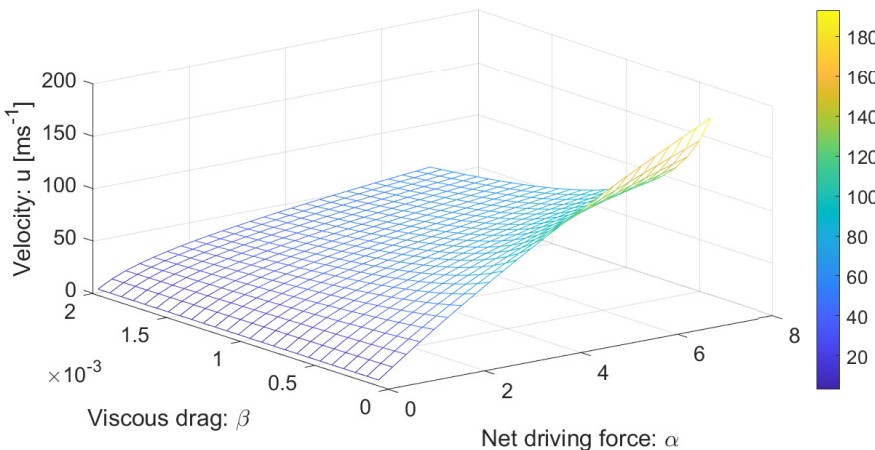

Figure 4: The influence of the model parameters $\alpha$ and $\beta$ on the landslide velocity. Colorbar shows velocity distributions in ms$^{-1}$.

For initial velocity less than $S_n$, i.e., $u_0 < S_n$, the landslide velocity increases rapidly just after its release, then ultimately (after a sufficiently long time) it approaches asymptotically to the steady state, $S_n$ (Fig. 2). This is the accelerating motion. On the other hand, if the initial velocity was higher than $S_n$, i.e., $u_0 > S_n$, the landslide velocity would decrease rapidly just after its release, then ultimately would asymptotically approaches to $S_n$. This is the decelerating motion (not shown here).

We have now two possibilities. First, we can describe $u(t; \alpha, \beta)$ as a function of time with $\alpha$, $\beta$ as parameters. This corresponds to the velocity profile of the particular landslide characterized by the geometrical, physical and mechanical parameters $\alpha$ and $\beta$ as time evolves. This has been shown in Fig. 2 for $u_0 < S_n$. A similar solution can be displayed for $u_0 > S_n$ for which the velocity would decrease and asymptotically approach to $S_n$.

### 3.2.4 Velocity described by the space of physical parameters

Second, we can investigate the control of the physical parameters on the landslide motion for a given time. This is achieved by plotting $u(\alpha, \beta; t)$ as a function of $\alpha$ and $\beta$, and considering time as a parameter. Figure 4 shows the influence of the parameters $\alpha$ and $\beta$ on the evolution of the velocity for a landslide motion for a typical time $t = 35$ s. The parameters $\alpha$ and $\beta$ enhance or control the landslide velocity completely differently. For a set of parameters $\{\alpha, \beta\}$, we can now provide an estimate of the landslide velocity. As mentioned earlier, the landslide velocity as high as 125 ms$^{-1}$ have been reported in the literature with their mean and common values in the range of 60 - 80 ms$^{-1}$ for rapid motions. This way, we can explicitly study the influence of the physical parameters on the dynamics of the velocity field and also determine their range of plausible values. This answers the question on how would the two similar looking, but physically differently characterized landslides move. They may behave completely differently.

### 3.2.5 A model for viscous drag

There exist explicit models for the interfacial drags between the particles and the fluid (Pudasaini, 2020) in the multiphase mixture flow (Pudasaini and Mergili, 2019). However, there exists no clear representation of the viscous drag coefficient for landslide which is the drag between the landslide and the environment. Often in applications, the drag coefficient ($\beta = C_{DV}$) is prescribed and is later calibrated with the numerical simulations to fit with the observation or data (Kattel et al., 2016; Mergili et al., 2020a, 2020b). Here, we explore an opportunity to investigate on how the characteristic landslide velocity (14) offers a unique possibility





to define the drag coefficient. Equation (14) can be written as

$$\beta = \frac{\alpha}{u_{max}^2}, \tag{16}$$

where, $u_{max}$ represents the maximum possible velocity during the motion as obtained from the (long-time)
steady-state behaviour of the landslide. Equation (16) provides a clear and novel definition (representation) of
the viscous drag in mass movement (flow) as the ratio of the applied forces to the square of the steady-state
(or a maximum possible) velocity the system can attain. With the representative mass $m$, (16) can be written
as

$$\beta = \frac{\frac{1}{2}m\alpha}{\frac{1}{2}mu_{max}^2}. \tag{17}$$

Equivalently, $\beta$ is the ratio between the one half of the "system-force", $\frac{1}{2}m\alpha$ (the driving force), and the
(maximum) kinetic energy, $\frac{1}{2}mu_{max}^2$, of the landslide. With the knowledge of the relevant maximum kinetic
energy of the landslide (Körner, 1980), the model (17) for the drag can be closed.

### 3.2.6 Landslide motion down the entire slope

Furthermore, we note that following the classical method by Voellmy (Voellmy, 1955) and extensions by Salm
(1966) and McClung (1983), the velocity models (8) and (11) can be used for multiple slope segments to
describe the accelerating and decelerating motions as well as the landslide run-out. These are also called the
release, track and run-out segments of the landslide, or avalanche (Gubler, 1989). However, for the gentle slope,
or the run-out, the frictional force may dominate gravity. In this situation, the sign of $\alpha$ in (5) changes. Then,
all the solutions derived above must be thoroughly re-visited with the initial condition for velocity being that
obtained from the lower end of the upstream segment. This way, we can apply the model (5) to analytically
describe the landslide motion for the entire slope, from its release, through the track and the run-out, as well
as to calculate the total travel distance. These methods can also be applied to the general solutions derived in
Section 4 and Section 5.

## 4 The Landslide Velocity: General Solution - I

For shallow motion the velocity may change locally, but the change in the landslide geometry may be param-
eterized. In such a situation, the force produced by the free-surface pressure gradient can be estimated. A
particular situation is the moving slab for which $h_g = 0$, otherwise $h_g \neq 0$. This justifies the physical signifi-
cance of (5).
The Lagrangian description of a landslide motion is easier. However, the Eulerian description provides a bet-
ter and more detailed picture of the landslide motion as it also includes the local deformation due to the
velocity gradient. So, here we consider the model equation (5). Without reduction, conceptually, this can
be viewed as an inviscid, non-homogeneous, dissipative Burgers' equation with a quadratic source of system
forces, and includes both the time and space dependencies of $u$. Exact analytical solutions for (5) can still
be constructed, however, in more sophisticated forms, and is very demanding mathematically. First, for the
notational convenience, we re-write (5) as:

$$\frac{\partial u}{\partial t} + g(u)\frac{\partial u}{\partial x} = f(u), \tag{18}$$

where, $g(u) = u$, and $f(u) = \alpha - \beta u^2$ correspond to our model (5). Here, $g$ and $f$ are sufficiently smooth
functions of $u$, the landslide velocity. Next, we construct exact analytical solution to the generic model (18).
For this, first we state the following theorem from Nadjafikhah (2009).
**Theorem 4.1:** *Let $f$ and $g$ be invertible real valued functions of real variables, $f$ is everywhere away from zero,*
*$\phi(u) = \int \frac{1}{f(u)} du$ is invertible, and $l(u) = \int \left( g\left(\phi^{-1}(u)\right)\right) du$. Then, $x = l(\phi(u)) + F\left[t - \phi(u)\right]$ is the solution*
*of (18), where $F$ is an arbitrary real valued smooth function of $t - \phi(u)$.*





Earth **Surface**
Dynamics
Discussions

To our problem (5), we have constructed the solution (below in Section 4.1), and reads as **(Solution D)**:

$$x = \frac{1}{\beta} \ln \left[ \cosh \left( \sqrt{\alpha\beta}\, \phi(u) \right) \right] + F\left[t - \phi(u)\right]; \quad \phi(u) = \frac{1}{2} \frac{1}{\sqrt{\alpha\beta}} \ln \left[ \frac{\sqrt{\alpha/\beta} + u}{\sqrt{\alpha/\beta} - u} \right]. \tag{19}$$

It is important to note, that in (19), the major role is played by the function $\phi$ that contains all the forces of
the system. Furthermore, the function $F$ includes the time-dependency of the solution. The amazing fact with
the solution (19) is that any smooth function $F$ with its argument $(t - \phi(u))$ is a valid solution of the model
equation. This means that, different landslides may be described by different $F$ functions. Alternatively, a
class of landslides might be represented by a particular function $F$. This is substantial.

### 4.1 Derivation of the solution to the general model equation

Here, we present the detailed derivation of the solution (19) to the landslide velocity equation (5). We derive
the functions $\phi$, $\phi^{-1}$, $l$ and $lo\phi$ that are involved in constructing the analytical solution in Theorem 4.1 for our
model (5). The first function $\phi$ is given by

$$\phi(u) = \int \frac{1}{f(u)} du = \int \frac{1}{\alpha - \beta u^2} du = \frac{1}{2\sqrt{\alpha\beta}} \ln \left[ \frac{\sqrt{\alpha/\beta} + u}{\sqrt{\alpha/\beta} - u} \right]. \tag{20}$$

With the substitution, $\tau = \phi(u)$ (which implies $u = \phi^{-1}(\tau)$), we obtain,

$$\phi^{-1}(\tau) = \sqrt{\frac{\alpha}{\beta}} \left[ \frac{\exp\left(2\sqrt{\alpha\beta}\,\tau\right) - 1}{\exp\left(2\sqrt{\alpha\beta}\,\tau\right) + 1} \right] = \sqrt{\frac{\alpha}{\beta}} \tanh \left( \sqrt{\alpha\beta}\,\tau \right). \tag{21}$$

So, now the second function $\phi^{-1}$ can be written in terms of $u$. However, we must be consistent with the physical
dimensions of the involved variables and functions. The quantities $u$, $\sqrt{\alpha\beta}$, $\sqrt{\alpha/\beta}$ and $\tau$ have dimensions of
$\text{ms}^{-1}$, $\text{s}^{-1}$, $\text{ms}^{-1}$ and s. Thus, for the dimensional consistency, the following mapping introduces a new multiplier
$\lambda$ with the dimension of $1/\text{ms}^{-2}$. Therefore, we have

$$\phi^{-1}(u) = \sqrt{\frac{\alpha}{\beta}} \tanh \left( \sqrt{\lambda\alpha\beta}\,u \right). \tag{22}$$

With this, the third function $l(u)$ yields:

$$l(u) = \int g\left( \phi^{-1}(u) \right) du = \int \phi^{-1}(u)\, du = \sqrt{\frac{\alpha}{\beta}} \int \tanh \left( \sqrt{\lambda\alpha\beta}\,u \right) du = \frac{1}{\lambda\beta} \ln \left[ \cosh \left( \lambda\sqrt{\alpha\beta}\,u \right) \right]. \tag{23}$$

The fourth function $l\left( \phi(u) \right) = (lo\phi)(u)$ is instantly achieved:

$$l\left( \phi(u) \right) = \left( \frac{\chi}{\lambda} \right) \frac{1}{\beta} \ln \left[ \cosh\left( \xi\lambda \right) \sqrt{\alpha\beta}\, \phi(u) \right], \tag{24}$$

where, as before, the multipliers $\chi$ and $\xi$ emerge due to the transformation and for the dimensional consistency,
they have the dimensions of $1/\text{ms}^{-2}$ and $\text{ms}^{-2}$, respectively. The nice thing about the groupings $(\chi/\lambda)$ and
$(\xi\lambda)$ is that they are now dimensionless and unity.
Utilizing these functions in Theorem 4.1, we finally constructed the exact analytical solution (19) to the model
equation (5) describing the temporal and spatial evolution of the landslide velocity.

### 4.2 Recovering the mass point motion

The amazing fact is that the newly constructed general analytical solution (19) is strong and includes both
the mass point solutions for velocity (11) and the position (13). Below we prove, that for a special choice of
the function $F$, (19) directly implies both (11) and (13). For this, consider a particular form of $F$ such that



$F(0) \equiv 0$, which is called a vacuum solution. First, $F(0) \equiv 0$ implies that $t = \phi(u)$. Then, with the functional
relation of $\phi(u)$ in (19), and after some simple algebraic operations, we obtain:

$$u = \sqrt{\frac{\alpha}{\beta}} \tanh \left[ \sqrt{\alpha\beta}\, t \right]. \tag{25}$$

Up to the constant of integration parameters (with $u_0 = 0$ at $t_0 = 0$), (25) is (11). So, the first assertion is
proved. Second, using $F(0) \equiv 0$ and $\phi(u) = t$ in (19), immediately yields

$$x = \frac{1}{\beta} \ln \left[ \cosh \left( \sqrt{\alpha\beta}\, t \right) \right]. \tag{26}$$

Again, up to the constant of integration parameters (with $x_0 = 0$, and $u_0 = 0$ at $t_0 = 0$), (26) is (13). This
proves the second assertion.
Moreover, we mention that (25) and (26) can also be obtained formally. This proves that the conditions used
on $F$ are legitimate. To see this, we differentiate (19) with respect to $t$ to yield

$$u = \frac{dx}{dt} = \sqrt{\frac{\alpha}{\beta}} \tanh \left[ \sqrt{\alpha\beta}\, \phi(u) \right] \frac{d\phi}{dt} + F'\left[ t - \phi(u) \right] \left( 1 - \frac{d\phi}{dt} \right). \tag{27}$$

But, differentiating $\phi$ in (19) with respect to $t$ and employing (10), we obtain $d\phi/dt = 1$, or $\phi = t$. Now, by
substituting these in (27) and (19) we respectively recover (25) and (26).
However, we note that $F$ in (19) is a general function. So, (19) provides a wide spectrum of analytical solutions
for the landslide velocity as a function of time and space, much wider than (11) and (13).

### 4.3  Some particular exact solutions

Here, we present some interesting particular exact solutions of (19) in the limit as $\beta \to 0$. For this purpose,
first we consider (5) with $\beta \to 0$, and introduce the new variables $\tilde{t} = \alpha t, \tilde{x} = \alpha x$. Then, (5) can be written as:

$$\frac{\partial u}{\partial \tilde{t}} + u \frac{\partial u}{\partial \tilde{x}} = 1. \tag{28}$$

Note that each term in this equation is dimensionless. We apply Theorem 4.1 and the underlying techniques to
(28). So, $f(u) = 1$ implies $\phi(u) = u, l(u) = u^2/2$, and $l(\phi(u)) = u^2/2$. Following the procedure as for (19), we
obtain the solution to (28) as: $\tilde{x} = \dfrac{u^2}{2} + F\left( \tilde{t} - u \right)$. However, the direct application of $\phi(u) = u$ in (19) leads
to the solution (that is more complex in its form): $\tilde{x} = \dfrac{1}{\beta} \ln \left[ \cosh \left( \sqrt{\beta}\, u \right) \right] + F\left( \tilde{t} - u \right)$. Then, in the limit, we
must have:

$$\lim_{\beta \to 0} \frac{1}{\beta} \ln \left[ \cosh \left( \sqrt{\beta}\, u \right) \right] = \frac{u^2}{2}. \tag{29}$$

This is an important mathematical identity we obtained as a direct consequence of Theorem 4.1 and (19).
Furthermore, the identity (29) when applied to (26) implies:

$$\lim_{\beta \to 0} x = \lim_{\beta \to 0} \frac{1}{\beta} \ln \left[ \cosh \left( \sqrt{\alpha\beta}\, t \right) \right] = \lim_{\beta \to 0} \frac{1}{\beta} \ln \left[ \cosh \left\{ \sqrt{\beta} \left( \sqrt{\alpha}\, t \right) \right\} \right] = \frac{1}{2} \alpha t^2. \tag{30}$$

Thus, $x = \frac{1}{2}\alpha t^2$, which is the travel distance in time when the viscous drag is absent. So, (29) is a physically
important identity.
Moreover, with the definition of $\tilde{x}$, for the particular choice of $F \equiv 0$, $\tilde{x} = \dfrac{u^2}{2} + F\left( \tilde{t} - u \right)$ results in $u(x; \alpha) =$
$\sqrt{2\alpha x}$, which is the solution given in (7). Furthermore, with the choice of $\tilde{x} = 0$, and $F = \tilde{t} - u$, we obtain
$u = 1 - \sqrt{1 - 2\alpha t}$, which for small $t$, can be approximated as $u \approx \alpha t$. But, in the limit as $\beta \to 0$, (11) brings
about $u = \alpha t$, which however, is valid for all $t$ values. Thus, (19) generalizes both solutions (7) and (11) in
numerous ways.


Earth **Surface**
Dynamics
Discussions

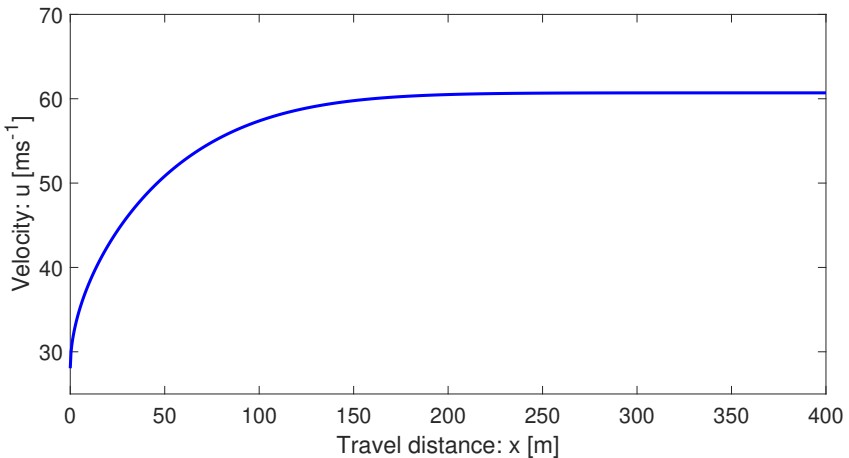

Figure 5: Velocity distribution given by (34).

## 4.4 Reduction to the classical Burgers' equation

Interestingly, by directly taking limit as $\beta \to 0$, from (19) we obtain

$$x = \frac{u^2}{2\alpha} + F\left(t - \frac{u}{\alpha}\right),\tag{31}$$

which can be written as

$$u^2 + 2\alpha\,F\left(t - \frac{u}{\alpha}\right) - 2\alpha\,x = 0.\tag{32}$$

Importantly, for any choice of the function $F$, (32) satisfies

$$\frac{\partial u}{\partial t} + u\frac{\partial u}{\partial x} = \alpha,\tag{33}$$

which reduces to the classical inviscid Burgers' equation when $\alpha \to 0$.

## 4.5 Some explicit expressions for $u$ in (19)

For a properly selected function $F$, (19) can be solved exactly for $u$. For example, consider a constant $F$, $F = \Lambda$. Then, an explicit exact solution is obtained as:

$$u = \sqrt{\frac{\alpha}{\beta}} \tanh\left[\frac{1}{2}\exp\left\{2\cosh^{-1}\left(\exp(\beta(x - \Lambda))\right)\right\}\right].\tag{34}$$

Figure 5 shows the velocity distribution given by (34) with $u \approx 28$ ms$^{-1}$ at $x = 0$ and $\Lambda = 0$, which reaches the steady-state at about $x = 150$ m, much faster than the solution given by (8) in Fig. 3.

However, other more general solutions could be found by considering different $F$ functions in (19). One such case is presented here. For the choice $F = \frac{1}{\beta}\ln\left[c\cosh\left\{\sqrt{\alpha\beta}(t - \phi(u))\right\}\right]$, where $c$ is a constant, (19) can be solved explicitly for $u$ in terms of $x$ and $t$, which, after lengthy algebra, takes the form:

$$u = \sqrt{\frac{\alpha}{\beta}} \tanh\left[\frac{1}{2}\left\{\cosh^{-1}\left(\frac{2}{c}\exp(\beta x) - \cosh\left(\sqrt{\alpha\beta}\,t\right)\right) + \sqrt{\alpha\beta}\,t\right\}\right].\tag{35}$$

The velocity profile along the slope as given by (35) is presented in Fig. 6 for $t = 1$ ms$^{-1}$ and $c = 1$. This solution is quite different to that in Fig. 3 produced by (8) which does not consider the local time variation of

Earth **Surface**
**Dynamics**
Discussions

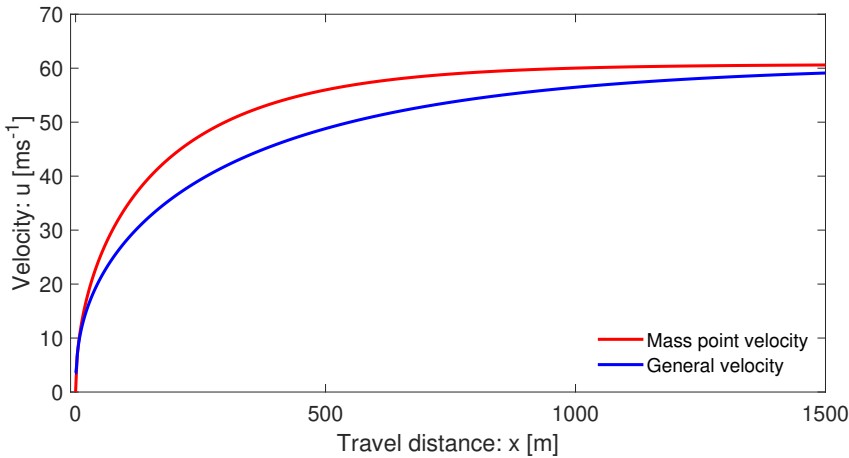

Figure 6: Evolution of the velocity field along the slope as given by (35) for general velocity against the mass point (or, center of mass) velocity corresponding to (8).

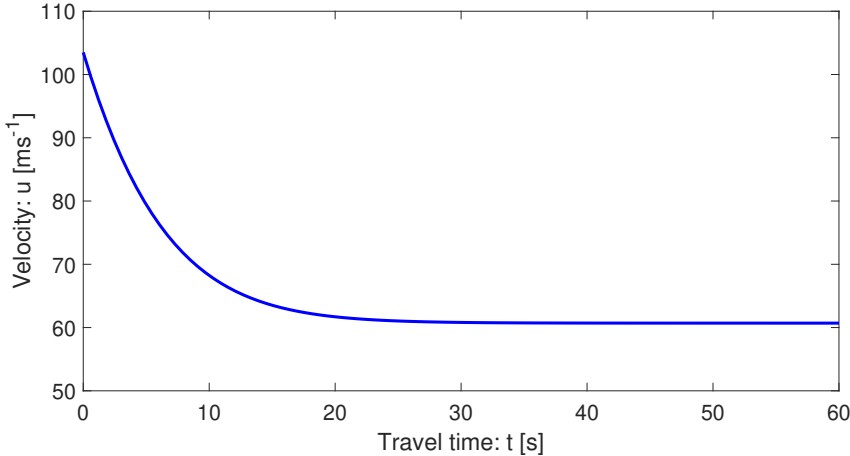

Figure 7: Time evolution of the velocity field as given by (35).

the velocity. From the dynamical perspective, the solution (35) is better than the mass point solution (8). The important observation is that the solution given by (8) substantially overestimates the legitimate more general solution (35) that includes both the time and space variation of the velocity field. The lower velocity with (35) corresponds to the energy consumption due to the deformation associated with the velocity gradient $\partial u/\partial x$ in (5). This will be discussed in more detail in Section 4.5 and Section 4.6.

Furthermore, Fig. 7 presents the time evolution of the velocity field given by (35) for $x = 25$ m, $c = -2$. This corresponds to the decelerating flow down the slope that starts with a very high velocity and finally asymptotically approaches to the steady-state velocity of the system. Similar situation has also been discussed at Section 3.2.3, but for a mass point motion.



Earth **Surface**
**Dynamics**
Discussions



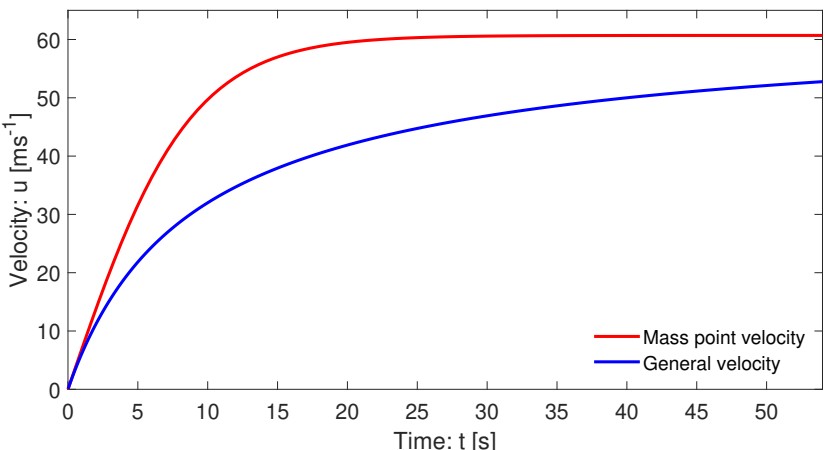

Figure 8: The velocity profiles for a landslide with the mass point motion as given by (11), and the motion including the internal deformation as given by the general solution (19). The two solutions behave fundamentally differently.

### 4.6 Description of the general velocity

A crucial aspect of a complex analytical solution is its proper interpretation. The general solution (19) can be plotted as a function of the travel distance $x$ and the travel time $t$. For the purpose of comparing the results with those derived previously, we select $F$ as: $F = [F_k(t - \phi(u))]^{p_w} + F_c$ with parameter values, $F_k = 5000, F_c = -500, p_w = 1/2$. Furthermore, $x$ is a parameter while plotting the velocity as a function of time. In these situations, in order to obtain physically plausible solution, the space parameter is selected as $x_0 = -600$. To match the origin of the mass point solution, in plotting, the time has been shifted by -2. Figure 8 depicts the two solutions given by (11) for the mass point motion, and the general solution given by (19) that also includes the internal deformation of the landslide associated with the velocity gradient or the non-linear advection $u\partial u/\partial x$ in (5). They behave essentially differently right after the mass release. The mass point model substantially overestimates landslide velocity derived by the more realistic general model.

### 4.7 A fundamentally new understanding

The new general solution (19) and its plot in Fig. 8 provides a fundamentally new aspect in our understanding of landslide velocity. The physics behind the substantially, but legitimately, reduced velocity provided by the general velocity (19) as compared to the mass point velocity (11) is revealed here for the first time. The gap between the two solutions increases steadily until a substantially large time (here about $t = 20$ s), then the gap is reduced slowly. This is so because, after $t = 20$ s the mass point velocity is close to its steady value (about 60.1 ms$^{-1}$). In the meantime, after $t = 20$ s, the general velocity continues to increase but slowly, and after a long time, it also tends to approach the steady-state. This substantially lower velocity in the general solution is realistic. Its mechanism can be explained. It becomes clear by analysing the form of the model equation (5). For the ease of analysis, we assume the accelerating flow down the slope. For such a situation, both $u$ and $\partial u/\partial x$ are positive, and thus, $u\partial u/\partial x > 0$. The model (5) can also be written as

$$\frac{\partial u}{\partial t} = \left(\alpha - \beta u^2\right) - u\frac{\partial u}{\partial x}. \tag{36}$$

Then, from the perspective of the time evolution of $u$, the last term on the right hand side can be interpreted as a negative force additional to the system (10) describing the mass point motion. This is responsible for the substantially reduced velocity profile given by (19) as compared to that given by (11). The lower velocity in





(19) can be perceived as the outcome of the energy consumed in the deformation of the landslide associated
with the spatial velocity gradient that can also be inferred by the negative force attached with $-u\partial u/\partial x$ in
(36). Moreover, $u\partial u/\partial x$ in (5) can be viewed as the inertial term of the system (Bertini et al., 1994). However,
after a sufficiently long time the drag is dominant, resulting in the decreased value of $\partial u/\partial x$. Then, the effect
of this negative force is reduced. Consequently, the difference between the mass point solution and the general
solution decreases. However, these statements must be further scrutinized.

## 5 The Landslide Velocity: General Solution - II

Below, we have constructed a further analytical solution to our velocity equation based on the method of
Montecinos (2015). Consider the model (5) and assign an initial condition:

$$\frac{\partial u}{\partial t} + u\frac{\partial u}{\partial x} = \alpha - \beta u^2, \ \ u(x,0) = s_0(x). \tag{37}$$

This is a non-linear advective - dissipative system, and can be perceived as an inviscid, dissipative, non-
homogeneous Burgers' equation. First, we note that, $H(x)$ is a primitive of a function $h(x)$ if $\dfrac{dH(x)}{dx} = h(x)$.
Then, we summarize the Montecinos (2015) solution method in a theorem:
**Theorem 5.1:** Let $\dfrac{1}{f(u)}$ be an integrable function. Then, there exists a function $\mathcal{E}\left(t, s_0(y)\right)$ with its primitive
$\mathcal{F}\left(t, s_0(y)\right)$, such that, the initial value problem

$$\frac{\partial u}{\partial t} + u\frac{\partial u}{\partial x} = f(u), \ \ u(x,0) = s_0(x), \tag{38}$$

has the exact solution $u(x,t) = \mathcal{E}\left(t, s_0(y)\right)$, where $y$ satisfies $x = y + \mathcal{F}\left(t, s_0(y)\right)$.
Following Theorem 5.1, after a bit lengthy calculation (below in Section 5.1), we obtain the exact solution
**(Solution E)** for (37):

$$u(x,t) = \sqrt{\frac{\alpha}{\beta}} \tanh\left[\sqrt{\alpha\beta}\, t + \tanh^{-1}\left\{\sqrt{\frac{\beta}{\alpha}} s_0(y)\right\}\right], \tag{39}$$

where $y = y(x,t)$ is given by

$$x = y + \frac{1}{\beta}\ln\left[\cosh\left\{\sqrt{\alpha\beta}\, t + \tanh^{-1}\left\{\sqrt{\frac{\beta}{\alpha}} s_0(y)\right\}\right\}\right] - \frac{1}{\beta}\ln\left[\cosh\left\{\tanh^{-1}\left\{\sqrt{\frac{\beta}{\alpha}} s_0(y)\right\}\right\}\right], \tag{40}$$

and, $s_0(x) = u(x,0)$ provides the functional relation for $s_0(y)$. In contrast to (19), (39)-(40) are the direct
generalizations of the mass point solutions given by (11) and (13). This is an advantage.
The solution strategy is as follows: Use the definition of $s_0(y)$ in (40). Then, solve for $y$. Go back to the
definition of $s_0(y)$ and put $y = y(x,t)$ in $s_0(y)$. This $s_0(y)$ is now a function of $x$ and $t$. Finally, put
$s_0(y) = f(x,t)$ in (39) to obtain the required general solution for $u(x,t)$. In principle, the system (39)-(40) may
be solved explicitly for a given initial condition. One of the main problems in solving (39)-(40) lies in inverting
(40) to acquire $y(x,t)$. Moreover, we note that, generally, (19) and (39)-(40) may provide different solutions.

### 5.1 Derivation of the solution to the general model equation

The solution method involves some sophisticated mathematical procedures. However, here we present a compact
but a quick solution description to our problem. The equivalent ordinary differential equation to the partial
differential equation system (37) is

$$\frac{d\hat{u}}{dt} = \alpha - \beta\hat{u}^2, \ \ \hat{u}(0) = s(0), \tag{41}$$





which has the solution

$$\hat{u}(t) = \mathcal{E}\left(t, s(0)\right) = \sqrt{\frac{\alpha}{\beta}} \tanh\left[\sqrt{\alpha\beta}\, t + \tanh^{-1}\left\{\sqrt{\frac{\beta}{\alpha}} s(0)\right\}\right]. \tag{42}$$

Consider a curve $x$ in the $x - t$ plane that satisfies the ordinary differential equation

$$\frac{dx}{dt} = \mathcal{E}\left(t, s_0(y)\right) = \sqrt{\frac{\alpha}{\beta}} \tanh\left[\sqrt{\alpha\beta}\, t + \tanh^{-1}\left\{\sqrt{\frac{\beta}{\alpha}} s_0(y)\right\}\right], \quad x(0) = y. \tag{43}$$

Solving the system (43), we obtain,

$$\begin{aligned}
x &= y + \mathcal{F}\left(t, s_0(y)\right) \\
&= y + \frac{1}{\beta}\ln\left[\cosh\left\{\sqrt{\alpha\beta}\, t + \tanh^{-1}\left\{\sqrt{\frac{\beta}{\alpha}} s_0(y)\right\}\right\}\right] - \frac{1}{\beta}\ln\left[\cosh\left\{\tanh^{-1}\left\{\sqrt{\frac{\beta}{\alpha}} s_0(y)\right\}\right\}\right]. \tag{44}
\end{aligned}$$

So, the exact solution to the problem (37) is given by

$$u(x,t) = \mathcal{E}\left(t, s_0(y)\right) = \sqrt{\frac{\alpha}{\beta}} \tanh\left[\sqrt{\alpha\beta}\, t + \tanh^{-1}\left\{\sqrt{\frac{\beta}{\alpha}} s_0(y)\right\}\right], \tag{45}$$

where $y$ satisfies (44).

## 5.2 Recovering the mass point motion

It is interesting to observe the structure of the solutions given by (39)-(40). For a constant initial condition, e.g., $s_0(x) = \lambda_0$, $s_0(y) = \lambda_0$, (39) and (40) are decoupled. Then, (39) reduces to

$$u(x,t) = \sqrt{\frac{\alpha}{\beta}} \tanh\left[\sqrt{\alpha\beta}\, t + \tanh^{-1}\left(\sqrt{\frac{\beta}{\alpha}}\lambda_0\right)\right]. \tag{46}$$

For $t = 0$, $u(x, 0) = u_0(x) = \lambda_0$, which is the initial condition. Furthermore, (40) takes the form:

$$x = x_0 + \frac{1}{\beta}\ln\left[\cosh\left\{\sqrt{\alpha\beta}\, t + \tanh^{-1}\left(\sqrt{\frac{\beta}{\alpha}}\lambda_0\right)\right\}\right] - \frac{1}{\beta}\ln\left[\cosh\left\{\tanh^{-1}\left(\sqrt{\frac{\beta}{\alpha}}\lambda_0\right)\right\}\right], \tag{47}$$

from which we see that for $t = 0$, $x = y = x_0$, which is the initial position. With this, we observe that (46) and (47) are the mass point solutions (11) and (13), respectively.

## 5.3 A particular solution

For the choice of the initial condition $s_0(x) = \sqrt{\frac{\alpha}{\beta}} \tanh\left[\cosh^{-1}\left\{\exp(\beta x)\right\}\right]$, combining (39) and (40), after a bit of algebra, leads to

$$u(x,t) = \sqrt{\frac{\alpha}{\beta}} \tanh\left[\cosh^{-1}\left\{\exp(\beta x)\right\}\right], \tag{48}$$

which, surprisingly, is the same as the initial condition. However, we can now legitimately compare (48) with the previously obtained solution (8), which is the steady-state motion with viscous drag. These two solutions have been presented in Fig. 9. The very interesting fact is that (8) and (48) turned out to be the same. For a real valued parameter $\beta$ and a real variable $x$, this reveals an important mathematical identity, that

$$\tanh\left[\cosh^{-1}\left\{\exp(\beta x)\right\}\right] = \sqrt{1 - \exp(-2\beta x)}. \tag{49}$$





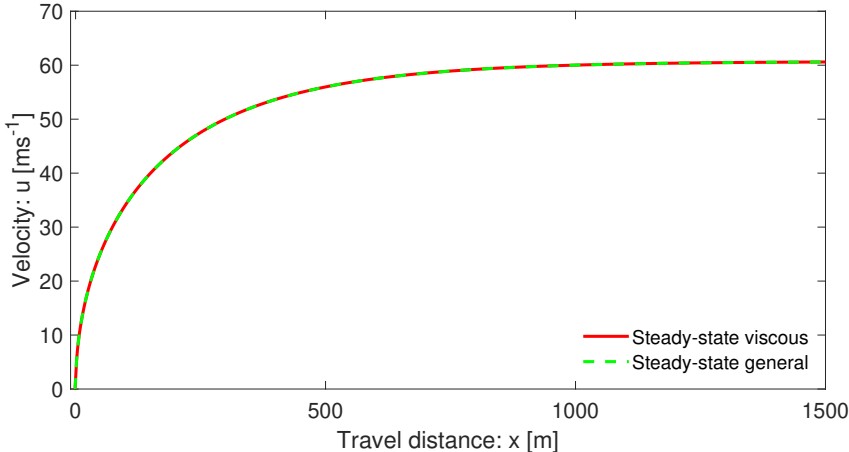

Figure 9: The velocity profile down a slope as a function of position for a landslide given by (39)-(40) reduced to the steady-state (48) against the steady-state solution with viscous drag given by (8). They match perfectly.

This means, the very complex function on the left hand side can be replaced by the much simpler function on the right hand side. Moreover, taking the limit as $\beta \to 0$ in (48) and comparing it with (7), we obtain another functional identity:

$$\lim_{\beta \to 0} \frac{1}{\sqrt{\beta}} \tanh\left[\cosh^{-1}\left\{\exp(\beta x)\right\}\right] = \sqrt{2x}. \tag{50}$$

These identities have mathematical significance.

## 5.4  Time marching general solution

Any initial condition can be applied to the solution system (39)-(40). For the purpose of demonstrating the functionality of this system, here we consider two initial conditions: $s_0(x) = x^{0.50}$ and $s_0(x) = x^{0.65}$. The corresponding results are presented in Fig. 10. This figure clearly shows time marching of the landslide motion that also stretches as it slides down. Such deformation of the landslide stems from the term $u\partial u/\partial x$ and the applied forces $\alpha - \beta u^2$ in our primary model (5). We will elaborate on this later. This proves our hypothesis on the importance of the non-linear advection and external forcing on the deformation and motion of the landslide. The mechanism and dynamics of the advection, stretching and approaching to the steady-state can be explained with reference to the general solution. For this, consider the lower panel with initial condition $s_0(x) = x^{0.65}$. At $t = 0.0$ s, (40) implies that $y = x$, then from (39), $u(x, t) = s_0(x)$, which is the initial condition. Such a velocity field can take place in relatively early stage of the developed motion of large natural events (Erismann and Abele, 2001; Huggel et al., 2005; Evans et al., 2009; Mergili et al., 2018). This is represented by the $t = 0.0$ s curve. For the next time, say $t = 2.0$ s, the spatial domain of $u$ expands and shifts to the right as defined by the rule (40). It has three effects in (39). First, due to the shift of the spatial domain, the velocity field $u$ is relocated to the right (down stream). Second, because of the increased $t$ value, and the spatial term associated with $\tanh^{-1}$, the velocity field is elevated. Third, as the tanh function defines the maximum value of $u$ (about 60.1 ms$^{-1}$), the velocity field is controlled (somehow appears to be rotated). This dynamics also applies for $t > 2.0$ s. These jointly produce beautiful spatio-temporal patterns in Fig. 10. Since the maximum of the initial velocity was already close to the steady-state value (the right-end of the curve), the front of the velocity field is automatically and strongly controlled, limiting its value to 60.1 ms$^{-1}$. So, although the rear velocity increases rapidly, the front velocity remains almost unchanged. After a sufficiently long time, $t \geq 15$ s, the rear velocity also approaches the steady-steady value. Then, the entire landslide moves downslope virtually with the constant steady-state velocity, without any substantial stretching. We can similarly describe the dynamics



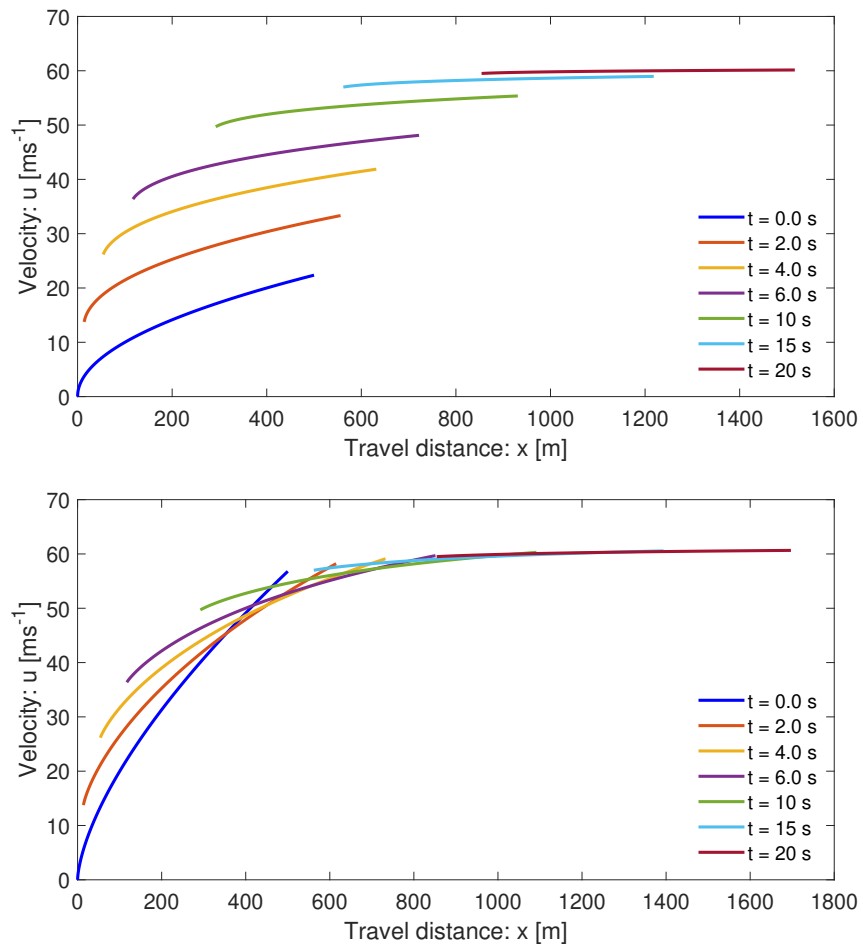

Figure 10: Time evolution of velocity profiles of propagating and stretching landslides down a slope, and as functions of position including the internal deformations as given by the general solution (39)-(40) of (5). The profiles correspond to the initial conditions $s_0(x) = x^{0.50}$ (top panel) and $s_0(x) = x^{0.65}$ (bottom panel), respectively.

for the upper panel in Fig. 10. However, these two panels reveal an important fact that the initial condition plays an important role in determining and controlling the landslide dynamics.

## 5.5 Landslide stretching

The stretching (or, deformation) of the landslide propagating down the slope depends on the evolution of its front and rear positions with maximum and minimum speeds, respectively. This has been shown in Fig. 11 corresponding to the initial condition $s_0(x) = x^{0.65}$ in Fig. 10. It is observed that the rear position evolves strongly non-linearly whereas the front position advances only weakly non-linearly.

In order to better understand the rate of stretching of the landslide, in Fig. 12, we also plot the difference between the front and rear positions as a function of time. It shows the stretching (rate) of the rapidly deforming landslide. The stretching dynamics is determined by the front and rear positions of the landslide in time, as has been shown in Fig. 11. In the early stages, the stretching increases rapidly. However, in later times (about





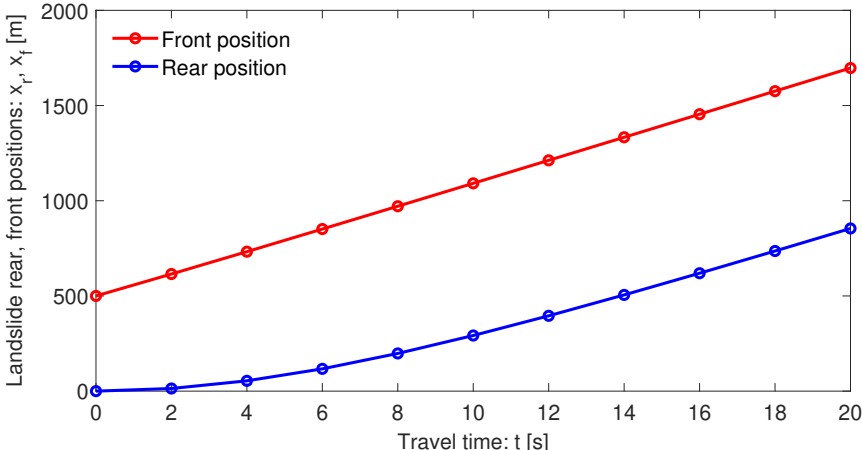

Figure 11: Time evolution of the front and rear positions of the landslide as it moves down the slope including the internal deformation given by the general solution (39)-(40) of (5), corresponding to the initial condition $s_0(x) = x^{0.65}$ in Fig. 10.

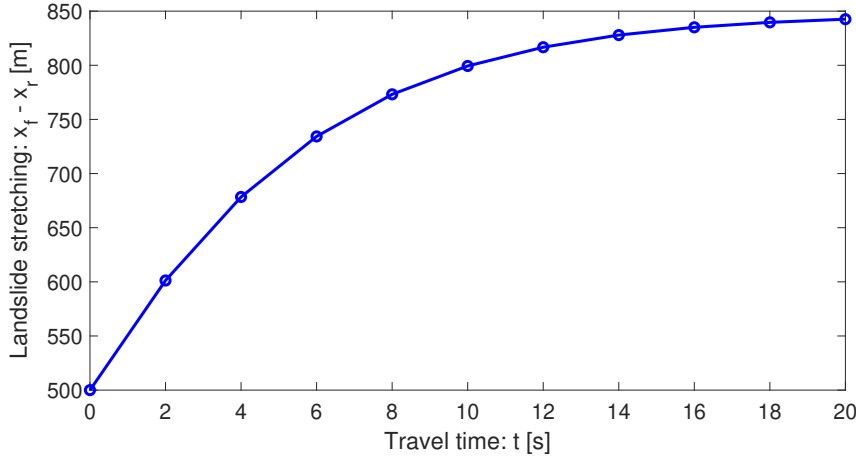

Figure 12: Time stretching of the landslide down the slope including the internal deformation given by the general solution (39)-(40) of (5), corresponding to the initial condition $s_0(x) = x^{0.65}$ in Fig. 10.

$t \geq 15$ s) it increases only slowly, and after a sufficiently long time, (the rate of) stretching vanishes as the landslide has already been fully stretched. This can be understood, because after a sufficiently long time, the motion is in steady-state. The two panels in Fig. 10 also clearly indicate that the stretching (rate) depends on the initial condition.

## 5.6 Describing the dynamics

The dynamics observed in Fig. 10 and Fig. 12 can be described with respect to the general model (5) or (37) and its solution given by (39)-(40). The nice thing about (39) is that it can be analyzed in three different ways: with respect to the first or second or both terms on the right hand side. If we disregard the first term involving time, then we explicitly see the effect of the second term that is responsible for the spatial variation





of $u$ for each time employed in (40). This results in the shift of the solution for $u$ to the right, and in the mean
time, the solution stretches but without changing the possible maximum value of $u$ (not shown). Stretching
continues for higher times, however, for a sufficiently long time, it remains virtually unchanged. On the other
hand, if we consider both the first and second terms on the right hand side of (39), but use the initial velocity
distribution only for a very small $x$ damain, say [0, 1], then, we effectively obtain the mass point solutions given
in Fig. 3 top and bottom panels corresponding to (8) and (11), respectively for the spatial and time evolutions
of $u$. This is so, because now the very small initial domain for $x$ essentially defines the velocity field as if it was
for a center of mass motion. Then, as time elapses, the domain shifts to the right and the velocity increases.
Now, plotting the velocity field as a function of space and time recovers the solutions in Fig. 3. In fact, if we
collect all the minimum values of $u$ (the left end points) in Fig. 10 (bottom panel) and plot them in space and
time, we acquire both the results in Fig. 3. These are effectively the mass point solutions for the spatial and
time variation of the velocity field, because these results only focus on the left end values of $u$, akin to the mass
point motion. This means, (40) together with (39) is responsible for the dynamics presented in Fig. 10, Fig. 11
and Fig. 12 corresponding to the term $u\partial u/\partial x$ and $\alpha - \beta u^2$ in the general model (5) or (37). So, the dynamics
is specially architectured by the advection $u\partial u/\partial x$ and controlled by the system forcing $\alpha - \beta u^2$, through the
model parameters $\alpha$ and $\beta$. This will be discussed in more detail in Section 5.7 - Section 5.9. This is a fantastic
situation, because, it reveals the fact that the shifting, stretching and lifting of the velocity field stems from
the term $u\partial u/\partial x$ in (37). After a long time, as drag strongly dominates the other system forces, the velocity
approaches the steady-state, practically the velocity gradient vanishes, and thus, the stretching ceases. Then,
the landslide just moves down the slope at a constant velocity without any further dynamical complication.

## 5.7 Rolling out the initial velocity

It is compelling to see how the solution system (39)-(40) rolls out an initially constant velocity across specific
curves. For this, consider an initial velocity $s_0(x) = 0$ in a small domain, say [0, 3], and take a point in it.
Then, generate solutions for different times, beginning with $t = 0.0$ s, with 2.0 s increments. As shown in
Fig. 13, the space and time evolutions of the velocity fields for a mass point motion given by (8) and (11)
have been exactly rolled-up and covered by the system (39)-(40) by transporting the initial velocity along these
curves (indicated by the star symbols). As explained earlier, the mechanism is such that, in time, (40) shifts
the solution point (domain) to the right and (39) up-lifts the velocity exactly lying on the mass point velocity
curves designed by (8) and (11). So, the system (39)-(40) generalizes the mass point motion in many different
ways.

## 5.8 Breaking wave and folding

Next, we show how the new model (5) and its solution system (39)-(40) can mould the breaking wave in
mass transport and describe the folding of a landslide. For this, consider a sufficiently smooth initial velocity
distribution given by $s_0(x) = 5\exp(-x^2/50)$. Such a distribution can be realized, e.g., as the landslide starts
to move, its center might have been moving at the maximum initial velocity due to some localized strength
weakening mechanism (examples include liquefaction, frictional strength loss; blasting; seismic shaking), and
the strength weakening diminishes quickly away from the center. This later leads to a highly stretchable
landslide from center to the back, while from center to the front, the landslide contracts strongly. The time
evolution of the solution has been presented in Fig. 14. The top panel for the usual drag as before ($\beta = 0.0019$),
while the bottom panel with higher drag ($\beta = 0.019$). The drag strongly controls the wave breaking and folding,
and also the magnitude of the landslide velocity. Here, we focus on the top panel, but similar analysis also
holds for the bottom panel.
Wave breaking and folding are often observed important dynamical aspects in mass transport and formation
of geological structures. Figure 14 reveals a thrilling dynamics. The most fascinating feature is the velocity
wave breaking and how this leads to the emergence of folding of the landslide. This can be explained with
respect to the mechanism associated with the solution system (39)-(40). As $u\partial u/\partial x$ is positive to the left
and negative to the right of the maximum initial velocity, the motion to the left of the maximum initial
velocity overtakes the velocity to the right of the maximum position. As the position of the maximum velocity





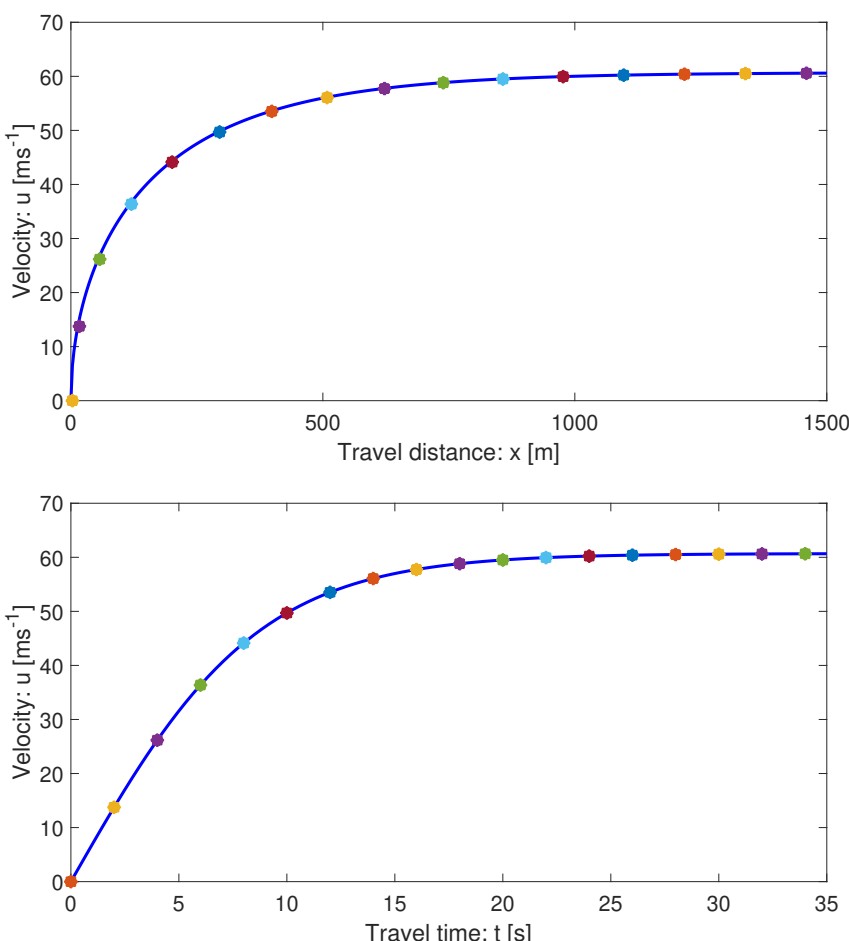

Figure 13: Spatial (top) and temporal (bottom) transportations of the initial velocity ($u = 0$) of the landslide down the slope by the general solution system (39)-(40) as indicated by the star markings for times $t = 0.0$ s, with 2.0 s increments. These solutions exactly fit with the space and time evolutions of the velocity fields for the mass point motions given by (8) and (11).

accelerates downslope with the fastest speed, after a sufficiently long time, a kink around the front of the
velocity wave develops, here after $t = 2$ s. This marks the velocity wave breaking (shock wave formation) and
the beginning of the folding. However, the rear stretches continuously. Although mathematically a folding may
refer to a singularity due to a multi-valued function, here we explain the folding dynamics as a phenomenon
that can appear in nature. In time, the folding intensifies, the folding length increases, but the folding gap
decreases. After a long time, virtually the folding gap vanishes and the landslide moves downslope at the
steady-state velocity with a perfect fold in the frontal part (not shown), while in the back, it maintains a
single large stretched layer. This happened collectively as the system (39)-(40) simultaneously introduced
three components of the landslide dynamics: downslope propagation, velocity up-lift and breaking or folding in
the frontal part while stretching in the rear. This physically and mathematically proves that the non-uniform
motion (with its maximum somewhere interior to the landslide) is the basic requirement for the development
of the breaking wave and the emergence of landslide folding. This is a seminal understanding.



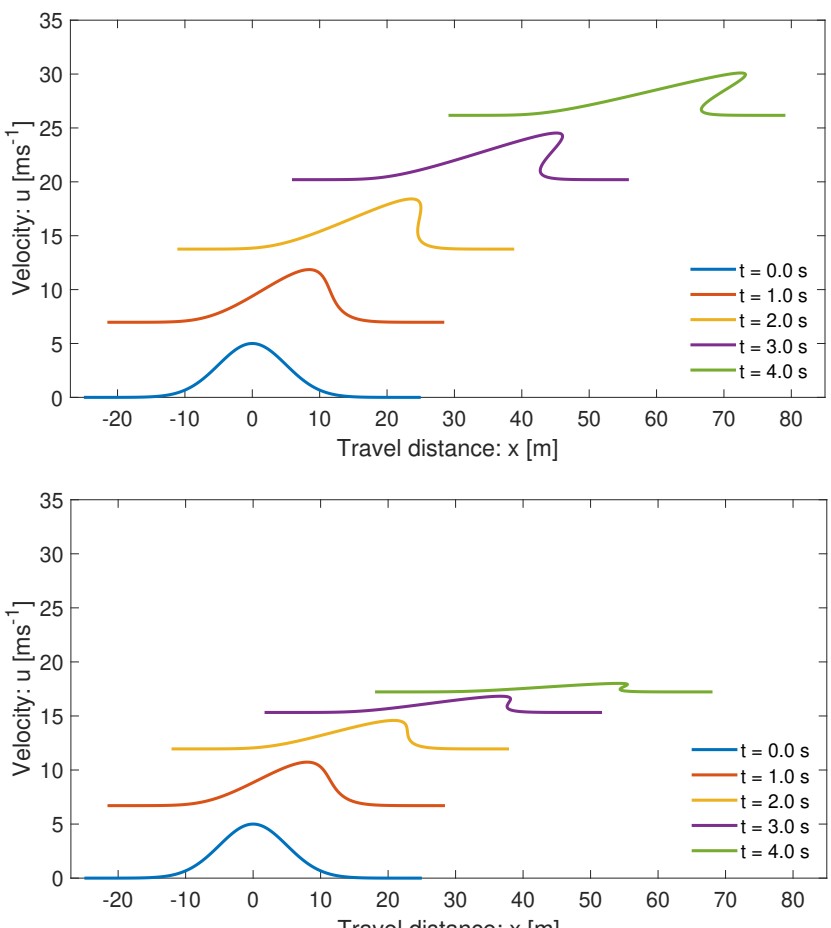

Figure 14: The breaking wave and folding as a landslide propagates down a slope. The top panel with drag $\beta = 0.0019$, while the bottom panel with higher drag, $\beta = 0.019$, which strongly controls the wave breaking and folding, and also the magnitude of the landslide velocity.

## 5.9 Recovering Burgers' model

As the external forcing vanishes, i.e., as $\alpha \to 0, \beta \to 0$, the landslide velocity equation (5) reduces to the classical inviscid Burgers' equation. Then, for $\alpha \to 0, \beta \to 0$, one would expect that the general solution (39)-(40) should also reduce to the formation of the shock wave and wave breaking generated by the inviscid Burgers' equation. In fact, as shown in Fig. 15, this has exactly happened. For this, the solution domain remains fixed, and the solution are not uplifted. This proves that Burgers' equation is a special case of our model (5).

## 5.10 The viscous drag effect

It is important to understand the dynamic control of the viscous drag on the landslide motion. For this, we set $\alpha \to 0$, but increased the value of the viscous drag parameter by one and two orders of magnitude. The results are shown in Fig. 16. In connection to Fig. 15, there are two important observations. First, the translation and stretching of the domain is solely dependent on the net driving force $\alpha$, and when it is set to zero, the domain remains fixed. Second, the viscous drag parameter $\beta$ effectively controls the magnitude of the velocity



Earth **Surface**
Dynamics
Discussions

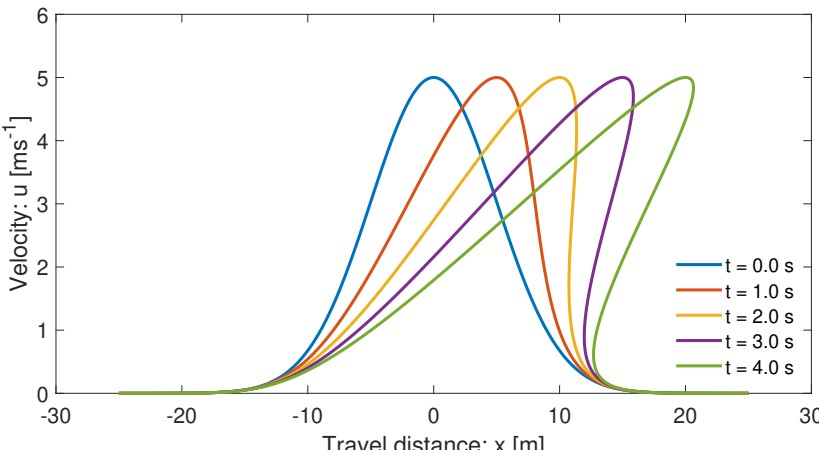

Figure 15: Recovering the Burgers' shock formation and breaking of the wave by the solution system (39)-(40) of the new model (5) in the limit of the vanishing external forcing, i.e., $\alpha \to 0, \beta \to 0$.

field and the wave breaking. Depending on the magnitude of the viscous drag coefficient, the generation of the shock wave and the wave breaking can be dampened (top panel) or fully controlled (bottom panel). The bottom panel further reveals, that with properly selected viscous drag coefficient, the new model can describe the deposition process of the mass transport and finally brings it to a standstill. In contrast to the classical inviscid Burgers' equation, due to the viscous drag effect, our model (5) is dissipative, and can be recognized as a dissipative inviscid Burgers' equation. However, here the dissipation is not due to the diffusion but due to the viscous drag.

# 6    Discussions

Analytical solutions of the underlying physical-mathematical models significantly improve our knowledge of the basic mechanism of the problem. On the one hand, exact, analytical solutions disclose many new and essential physics, and thus, may find applications broadly in environmental and engineering mass transport down natural slopes or industrial channels. The reduced and problem-specific solutions provide important insights into the full behavior of the complex landslide system, mainly the landslide motion with non-linear internal deformation together with the external forcing. On the other hand, exact analytical solutions to simplified cases of non-linear model equations are necessary to calibrate numerical simulations (Chalfen and Niemiec, 1986; Pudasaini, 2011, 2016; Ghosh Hajra et al., 2018). For this reason, this paper is mainly concerned about the development of a new general landslide velocity model and construction of several novel exact analytical solutions for landslide velocity.

Analytical solutions provide the fastest, cheapest, and probably the best solution to a problem as measured from their rigorous nature and representation of the dynamics. Proper knowledge of the landslide velocity is required in accurately determining the dynamics, travel distance and enormous destructive impact energy carried by the landslide. The velocity of a landslide is associated with its internal deformation (inertia) and the externally applied system forces. The existing influential analytical landslide velocity models do not include many important forces and internal deformation. The classical analytical representation of the landslide velocity appear to be incomplete and restricted, both from the physics and the dynamics point of view. No velocity model has been presented yet that simultaneously incorporates inertia and the externally applied system forces that play crucial role in explaining important aspects of landslide propagation, motion and deformation.

We have presented the first-ever, analytically constructed simple, but more general landslide velocity model.



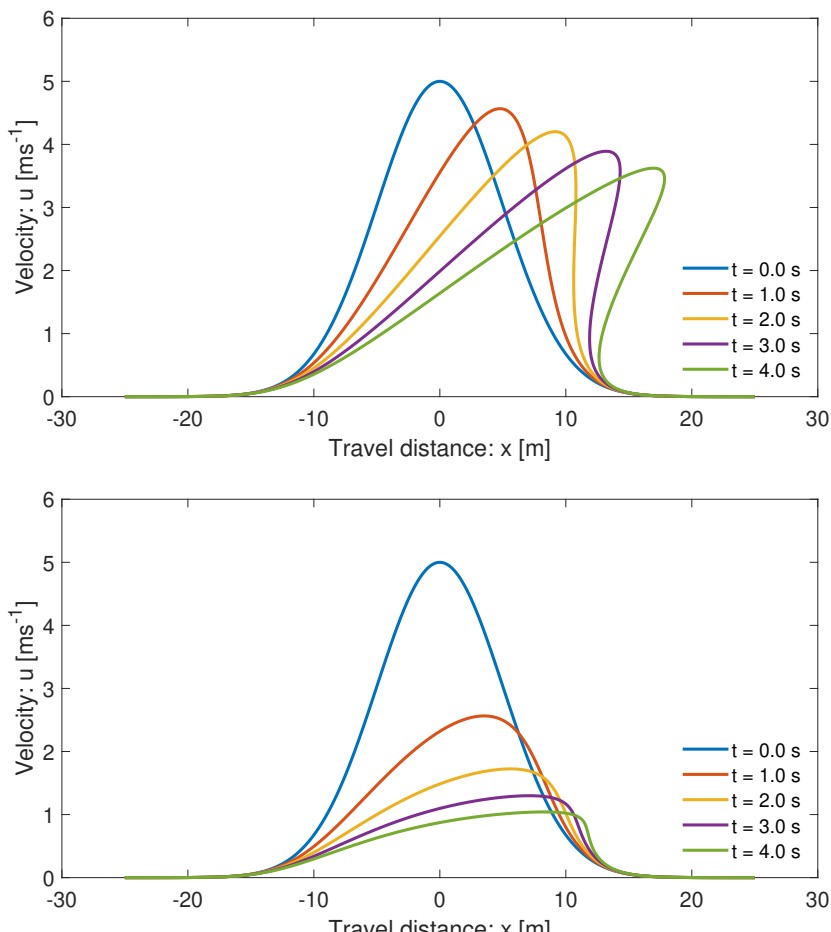

Figure 16: The control of the viscous drag on the dynamics of the landslide. The net driving force is set to zero, i.e., $\alpha = 0$. The viscous drag has been amplified by one and two orders of magnitudes in the top ($\beta = 0.019$) and bottom ($\beta = 0.19$) panels, showing dampened or complete prevention of shock formation and wave breaking, respectively.

There are two main collective model parameters: the net driving force and drag. By rigorous derivations of the exact analytical solutions, we showed that incorporation of the non-linear advection and external forcing is essential for the physically correct description of the landslide velocity. In this regard, we have presented a novel dynamical model for landslide velocity that precisely explains both the deformation and motion by quantifying the effect of non-linear advection and the system forces.

Different exact analytical solutions for landslide velocity constructed in this paper independently support each other and are compatible with the physics of landslide motion. These physically meaningful solutions can potentially be applied to calculate the complex non-linear velocity distribution of the landslide. Our new results reveal that solutions to the more general equation for the landslide motion are wide-ranging and include the classical mass point Voellmy and Burgers models for mass transport as special cases.

The new landslide velocity model and and its advanced exact solutions made it possible now to analytically study the complex landslide dynamics, including non-linear propagation, stretching, wave breaking and folding. Moreover, these results clearly indicate that the proper knowledge of the model parameters $\alpha$ and $\beta$ is crucial





in reliable prediction of the landslide dynamics.

## 6.1 Advantages of the new model and its solutions

The new model may describe the complex dynamics of many extended physical and engineering problems
appearing in nature, science and technology - connecting different types of complex mass movements and
deformations. Specifically, the advantage of the new model equation is that the more general landslide velocity
can now be obtained explicitly and analytically, that is very useful in solving relevant engineering and applied
problems and has enormous application potential. Broadly speaking, this is the first-ever physics-based model
to do so.
There are three distinct situations in modelling the landslide motion: (i) The spatial variation of the flow
geometry and velocity can be negligible for which the entire landslide effectively moves as a mass point without
any local deformation. This refers to the classical Voellmy model. (ii) The geometric deformation of the
landslide can be parameterized or neglected, however, the spatial variation of the velocity field may play a
crucial role in the landslide motion. In this circumstance, the landslide motion can legitimately be explained
by the full form of the new landslide velocity equation (5). The constructed general solutions (19) and (39) -
(40) of this model have revealed many important features of the dynamically deforming and advecting landslide
motions. (iii) Both the landslide geometry and velocity may substantially change locally. Then, no assumptions
on the spatial gradient of the geometry and velocity can be made. For this, only the full set of the basic model
equations (1) - (2) can explain the landslide motion. While models and simulation techniques for situations (i)
and (iii) are available in the literature, (ii) is entirely new, both physically and mathematically. It is evident
that dynamically (ii) plays an important role, first in making the bridge between the two limiting solutions, and
second, by providing the fastest, cheapest and the most efficient solution of the underlying problem. Solutions
(19) and (39)-(40) include the local deformation associated with the velocity gradient. However, except for
parameterization, (19) and (39)-(40) do not explicitly include the geometrical deformation. As long as the
spatial change in the landslide geometry is insignificant, we can use (19) or (39)-(40) to describe the landslide
motion. These solutions also include mass point motions, and are valid before the fragmentation and/or the
significant to large large geometric deformations. However, when the geometric deformations are significant,
we must use (1) and (2) and solve them numerically with some high resolution numerical methods (Tai et al.,
2002; Mergili et al., 2017, 2020a,b).
The model (19) or (39)-(40) and (1)-(2) are compatible and can be directly coupled. Such a coupling between
the geometrically negligibly- or slowly- deforming landslide motion described by (19) or (39)-(40) and the full
dynamical solution with any large to catastrophic deformations described by (1)-(2) is novel. First, this allows
us to consistently couple the negligible or slowly deformable landslide with a fast (or, rapidly) deformable
flow-type landslide (or, debris flow). Second, our method provides a very efficient simulation due to instant
exact solution given by (19) or (39)-(40) prior to the large external geometric deformation that is then linked
to the full model equations (1)-(2). The computational software such as r.avaflow (Mergili et al., 2017, 2020a,
2020b; Pudasaini and Mergili, 2019) can substantially benefit from such a coupled solution method. Third,
importantly, this coupling is valid for single-phase or multi-phase flows, because the corresponding model (5)
is derived by reducing the multi-phase mass flow model (Pudasaini and Mergili, 2019).
Burgers' equation has no external forcing term. The solution domain remains fixed and does not stretch and
propagate downslope. So, the initial velocity profile deforms and the wave breaks within the fixed domain.
In contrast, our model (5) is fundamentally characterized and explained simultaneously by the non-linear
advection $u\partial u/\partial x$ and external forcing, $\alpha - \beta u^2$. The first designs the main dynamic feature of the wave, while
the later induces rapid downslope propagation, stretching of the wave domain and quantification of the wave
form and magnitude. These special features of our model are often observed phenomena in mass transport,
and are freshly revealed here.





## 6.2 Compatibility, reliability and generality of the solutions

Within their scopes and structures, many of the analytical solutions constructed in Sections 3 - 5 are similar. This effectively implies the physical aspects of our general landslide velocity model (5), and also the compatibility and reliability of all the solutions. We have seen that the solutions (19) and (39)-(40) recover all the mass point motions given by (11) and (13). However, the analyses presented in Sections 3 - 5 reveal that from the physical and dynamical point of view, the velocity profiles given by (19) and (39)-(40) as solutions of the general model for the landslide velocity (5) are much wider and better than those given by (11) and (13) as solutions of the mass point model (10).

Structurally, the solutions presented in Section 3 are only partly new, yet they are physically substantially advanced. However, in Section 4 and 5 we have presented entirely novel solutions, both physically and structurally. From physical and mathematically point of view, particularly important is the form of the general velocity model (5). First, it extends the classical Voellmy mass point model (Voellmy, 1955) by including: (i) much wider physical aspects of landslide types and motions, and (ii) the landslide dynamics associated with the internal deformation as described by the spatial velocity gradient associated with the advection. Second, the model (5) is the direct extension of the inviscid Burgers' equation by including a (quadratic) non-linear source as a function of the state variable. This source term contains all the applied forces appearing from the physics and mechanics of the landslide motion.

Moreover, as viewed from the general structure of the model (5), all the solutions constructed here can be utilized for any physical problems that can be cast and represented in the form (5), but independent of the definition of the model parameters $\alpha$ and $\beta$. These parameters, and the initial (or, boundary) condition are dependent on the physics of the problem under consideration.

## 6.3 Importance and implications

A further important feature is the construction of the general and particular exact analytical solutions to the model (5) and the description of their physical significance and application in quickly and efficiently solving technical problems. So, in short, the new model (5) and its solutions have broad implications, mathematically, physically and technically.

By deriving a general landslide velocity model and its various analytical exact solutions, we made a breakthrough in correctly determining the velocity of a deformable landslide that is controlled by several applied forces as it propagates down the slope. We achieve a novel understanding that the inertia and the forcing terms ultimately regulate the landslide motion and provide physically more appropriate analytical description of landslide velocity, dynamic impact and inundation. This addresses the long-standing scientific question of explicit and full analytical representation of velocity of deformable landslides. Such a description of the state of landslide velocity is innovative.

As the analytically obtained values well represent the velocity of natural landslides, technically, this provides a very important tool for the landslide engineers and practitioners in quickly and accurately determining the landslide velocity. The general solutions presented here reveal an important fact that accurate information about the mechanical parameters, state of the motion and the initial condition is very important for the proper description of the landslide motion. We have extracted some interesting particular exact solutions from the general solutions. As direct consequences of the new general solutions, some important and non-trivial mathematical identities have been established that replace very complex expressions by straightforward functions.

## 7 Summary

While existing analytical landslide velocity models cannot deal with the internal deformation and mostly fail to integrate a wide spectrum of externally applied forces, we developed a simple but general analytical model that is capable of including both of these important aspects. In this paper, we (i) derived a general landslide velocity model applicable to different types of landslide motions with internal deformations, and (ii) solve it analytically





to obtain several exact solutions as a function of space and time for landslide motion, and highlight the essence of the new model to enhance our understanding of landslide dynamics. The model is developed by reducing a multi-phase mass flow equations (Pudasaini and Mergili, 2019) and includes the internal (local) deformation due to non-linear advection (inertia), and the external forcing consisting of the extensive net driving force and viscous drag. The model describes a dissipative system and involves dynamic interactions between the advection and external forcing that control the landslide deformation and motion. In the form, our model constitutes a unique and new class of non-linear advective - dissipative system with quadratic external forcing as a function of state variable, containing all system forces. The new model is a more general formulation, but can also be viewed as an extended inviscid, non-homogeneous, dissipative Burgers' equation. The form of the new equation is important as it may describe the dynamical state of many extended physical and engineering problems appearing in nature, science and technology. From the physical and mathematical point of view, there are two crucial novel aspects: First, it extends the classical Voellmy model due to the broad physics carried by the model parameters and additionally explains the dynamics of deforming landslide described by advection. So, our model provides a better and more detailed picture of the landslide motion by including the local deformation. Second, it extends the classical inviscid Burgers' equation by including the non-linear source term, as a quadratic function of the field variable. The source term accommodates the mechanics of underlying problem through the physical parameters, the net driving force and the dissipative viscous drag.

Due to the non-linear advection and quadratic forcing, the new general landslide velocity model poses a great mathematical challenge to derive explicit analytical solutions. We focused on constructing several new and general exact analytical solutions in more sophisticated forms. These solutions are strong, recover all the mass point motions and provide much wider spectrum for the landslide velocity than the classical Voellmy and Burgers' solutions. We have illustrated that the new system of solutions generalize the mass point motion in many different ways. The major role is played by the non-linear advection and system forces. The general solutions provide essentially new aspects in our understanding of landslide velocity. We have analytically proven that after a sufficiently long distance or time, the net driving force and drag always maintain a balance, resulting in the terminal velocity. We have also presented a new model for the viscous drag as the ratio between one half of the system-force and the relevant kinetic energy.

With the general solution, we revealed that different classes of landslides can be represented by different solutions under the roof of one velocity model. General solutions allowed us to simulate the progression and stretching (deforming) of the landslide as it slides down. Such deformation stems from the non-linear advection in our primary model. This proves our hypothesis on the importance of advection term on the deformation and motion. The mechanisms of advection, stretching and approaching to the steady-state have been explained with reference to the general solution. We disclose the fact that the shifting and stretching of the velocity field stem from the external forcing and non-linear advection. Also after a long time, as drag strongly dominates the system forces, the velocity approaches the steady-state, practically the velocity gradient vanishes, and thus, the stretching ceases. Then, the landslide propagates down the slope just at a constant velocity.

We have shown, that the general solution system can generate complex breaking waves in advective mass transport and describe the folding process of a landslide. Such phenomena have been presented and described mechanically for the first-time. The most fascinating feature is the dynamics of the wave breaking and the emergence of folding. These have been explained with respect to the intrinsic mechanism of our solution. This happened collectively as the solution system simultaneously introduces three important components of the landslide dynamics: downslope propagation and stretching of the domain, velocity up-lift, and breaking or folding in the frontal part while stretching in the rear. This physically proves that the non-uniform motion is the basic requirement for the development of breaking wave and emergence of the landslide folding. This is a novel understanding. We disclosed the fact that the translation and stretching of the domain, and lifting of the velocity field solely depends on the net driving force. Similarly, the viscous drag fully controls the shock wave generation, wave breaking and folding, and also the magnitude of the landslide velocity. Furthermore, with properly selected system force and viscous drag, the new model can describe the deposition or the halting process of the mass transport. As the external forcing vanishes, general solutions automatically reduce to the classical shock wave generated by the inviscid Burgers' equation but without domain translation, stretching



and lifting. So, in contrast to the classical inviscid Burgers' equation, due to the viscous drag, our model is dissipative. This proves that the inviscid Burgers' equation is a special case of our general model. The theoretically obtained velocities are close to the often observed values in natural events including landslides and debris avalanches. This indicates the broad application potential of the new landslide velocity model and its exact analytical solutions in quickly solving engineering and technical problems in accurately estimating the impact force that is very important in delineating hazard zones and for the mitigation of landslide hazards.

# Acknowledgements

Shiva P. Pudasaini acknowledges the financial support provided by the Technical University of Munich with the Visiting Professorship Program, and the international research project: AlpSenseRely − Alpine remote sensing of climate-induced natural hazards - from the Bayerisches Staatsministerium für Umwelt und Verbraucher-schutz, Munich, Bayern.

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
