# Peer review of "The Landslide Velocity"

_Earth Surface Dynamics, 2021_

## Author Comment (AC1)

**Response to Reviewer-1: MS #: eSurf-2021-81**

Reviewer's comments are denoted by C and our responses are denoted by R, respectively.

**General comments**

C: This manuscript presented a simple and physics-based general analytical landslide velocity model which helps solve more landslide problems. The logic of the manuscript is clear, and the structure is reasonable, but there are still some problems.

R: We very much appreciate the Reviewer for supporting our work. Our sincere thanks to the Reviewer for your time and constructive comments and explicit suggestions that results in the substantially improved manuscript in which we appropriately address all the concerns you raised.

**Specific comments**

C1.: In my opinion, the title of this manuscript should be modified. You can make it more specific. It can't be seen from the current title that what you introduce in your article is related to landslide speed model.

R1.: Thank you very much for the suggestion. We can understand the reviewer's concern. We also thought to change the title to "A novel class of non-linear advective - dissipative system". We can be open to this choice. There are two major aspects of this manuscript. First, the development of the new Landslide Velocity equation (5), which, based on the physical parameters, involved forces and the dynamics, namely, the net driving and the resisting forces, presents a novel class of non-linear advective - dissipative system, the physical-mathematical model for landslide velocity. This has been exclusively discussed in Section 2.3. Second, construction of several novel exact analytical solutions to the model (5) for the velocity of the landslide. So, the overall essence of the manuscript is on the landslide velocity. The nice thing is that the same equation (5) can describe many different natural and physical phenomena by appropriately changing $u$ (the velocity) to any relevant state variable. Even the simplified version of (5); the equation (6) or (10); can describe wide range of physical phenomena, including: the Schrödinger equation, the Ermakov-Pinney, and the Friedmann equations in physical cosmology (https://arxiv.org/pdf/2112.11526.pdf). However, since our principle model is developed for velocity, and the manuscript is in the earth science journal, we think that the present title of the manuscript fits very well to what it describes. Also, the title as it stands now is nice. Whatever we call it, scientific communities may use the general exact analytical solutions constructed here to the context it fits to their interests.

C2.: Figure 3 is the combination of Figure 1 and Figure 2. There is no need to draw it again.

R2.: Thanks a lot for this legitimate suggestion. Fig. 3 will be removed, and the text improved accordingly.

C3.: The conclusion could have been a little more concise and organized. It can be divided into 1, 2 and 3 points.

R3.: We can make the Conclusion [Summary] much more concise and reduce it substantially by focusing only to the major outcomes. However, we think, it looks nicer in a plain text without dividing into points.

C4.: In Fig.10, Which graph represents the initial conditions $s_0(x) = x^{0.50}$ (top panel) or $s_0(x) = x^{0.65}$ (bottom panel)? In Fig.14, Which graph represents the initial conditions $\beta = 0.0019$ or $\beta = 0.019$? The different initial conditions should be represented in the diagram so that we can quickly distinguish between them.

R4.: The caption of Fig.10 will be improved, where, "The profiles correspond to the initial conditions $s_0(x) = x^{0.50}$ (top panel) and $s_0(x) = x^{0.65}$ (bottom panel), respectively." will read "The profiles evolve based on the initial conditions $s_0(x) = x^{0.50}$ (top panel, at $t = 0.0$ s) and $s_0(x) = x^{0.65}$ (bottom panel, at $t = 0.0$ s), respectively." In Fig.14, $\beta = 0.0019$ and $\beta = 0.019$ will be placed in the top and bottom figure panels, respectively. And, the figure caption will be improved consistently, from "The top panel with drag $\beta = 0.0019$, while the bottom panel with higher drag, $\beta = 0.019$, which strongly controls the wave breaking and folding,

and also the magnitude of the landslide velocity." to "The top panel with lower drag, while the bottom panel with higher drag, showing the drag strongly controls the wave breaking and folding, and also the magnitude of the landslide velocity."

C5.: In section 6.1: "the significant to large large geometric deformations" is not written correctly?

R5.: We change it to: "the significant to large geometric deformations"

---

## Author Comment (AC2)

**Response to Reviewer-2: MS #: eSurf-2021-81**

Reviewer's comments are denoted by C and our responses are denoted by R, respectively.

**General comments**

C. The paper by Pudasaini & Krautblatter presents a series of detailed analytical solutions for the calculation of the velocity of a landslide, by considering the depth-averaged forms of incompressible mass and momentum conservation equations for a mixture of solid materials and a liquid (treated as a single-phase flowing material) as a starting point. It is shown that the analytic solutions though they are simplifications of the reality can capture many crucial mechanisms, such as the longitudinal stretching (the model remains 2D) of the mass during its movement along the slope, wave breaking and landslide folding. Moreover, simplified versions of the new analytic equations (equations numbered 11, 19 and 39-40 in the manuscript) allow to retrieve other simplified existing models, such as the ones given by solving the center of mass model developed by Voellmy –initially for snow avalanches– or the inviscid Burgers equations developed in fluid mechanics.

I found the scientific content of the paper very interesting with many interesting ideas discussed all along the development of these new analytic equations for the velocity of a landslide. Moreover I think that the paper is a niece piece of work for educational purposes, though I have some concerns regarding the presentation (see my general recommendation below).

The equations proposed by Pudasaini and Krautblatter remain a simplification of the reality but they take into account several key contributions to describe the motion of a landslide and thus are powerful tools to assess landslide risk and conduct relevant calculations for practitioners who are in charge of the hazard quantification and mitigation. They can complement more sophisticated calculations based on complicated numerical simulations, better than current analytic solutions based on center of mass models largely used in engineering (Voellmy-type models) can do.

As such, I have a very positive feeling on this work. However, I have a number of concerns that need to be addressed before I can definitely recommend it for publication.

The scientific content is very good and very interesting, overall. My concerns are on the presentation of the material. I must say that the paper was very difficult to digest (though I was excited to read it!). I really think the authors need to revise a lot the presentation to make the paper shorter and clearer.

R: We very much appreciate the Reviewer for supporting our work. Our sincere thanks to the Reviewer for your time and constructive comments and explicit suggestions that results in the substantially improved manuscript in which we appropriately address all the concerns you raised, including the presentation, length and the clarity of the paper. In the original submission, we thought we appropriately choose some words and phrases that would fit with the spirit of the text and would pleasure the readers, which however, will now be removed or made milder according to the reviewer's suggestions.

C: Here are some suggestions for that purpose:

C 1): the introduction is very very long: it really needs to be revised and shortened. There are a number of arguments repeated. I don't think that the authors need to oversell their results. They just need to present the facts and summarize them, this will be more efficient.

R 1): Thank you very much for this very reasonable suggestion. The Introduction was about one and a half page long in the initial submission, which we thought was ok for a professional paper. We do not intend to overvalue the work, but realize we need to considerably improve the presentation. However, we agree with the reviewer that there are number of statements repeated. Following your suggestion, we will present the facts and summarize them in a much more efficient and concise way resulting in substantially reduced Introduction.

C 2): I would suggest the authors to change the organization of the paper by starting with a presentation of

the new analytic solutions: firstly with section 3.2 (eq 11), secondly with section 4 (eq. 19) and thirdly with section 5 (eqs 39-40). The simpler solutions for steady-state motion (eqs. 6, 7, 8) can be shortly presented as specific cases, after eq 11 or even in a supplementary material. The manuscript needs to be shortened.

R 2): We understand the reviewer's idea of first presenting the main general solutions, then collect all simpler solutions, or put them in a supplementary. We appreciate for this suggestion. However, the spirit of the ms is first to present simple solutions, similar to those available in literature, that many readers may be acquainted with. Then, tell what was not included or was not possible in the existing models, solutions and approaches, what we need to generalize and how we can achieve them. We presented the general solutions, then showed how those solutions can do more than existing ones, while in the mean time we analytically proved that our general solutions can directly recover all existing solutions as special cases. We think this way Sections 3-5 are well structured, smooth, easy to follow, also well formulated for educational uses. We hope the readers will like it. However, we have now identified many places where we can still substantially reduce Sections 3-5. We ask for the reviewer's support for this.

C 3): the summary section (section 7) is very long, with many repetitions again. And most of the material is already clear and said in the discussion section (section 6) or elsewhere in the paper. I think the summary section should be removed. If there are key ideas in the section 7 which the authors want to keep, they should be moved to the section 6 then and that's it.

R 3): We agree that there are many repetitions. In revising the ms, we remove all repetitions from Section 6, reducing it substantially. We will clearly re-write Section 6 to discuss the main results, while Section 7 will summarize the main findings, but without overlap. Moreover, we will largely reduce Section 7, making it much concise. This way it will be better.

C 4): there many superlatives or sentences that intend to praise the work done by the authors. Please let the readers themselves appreciate the quality of your work! I think those superlatives or statements that oversell your work are not needed and should be removed: see my suggestions in the list of specific comments below.

R 4): Some specific words were used to make it clearer the importance of the new findings as they are from the physical and mathematical ground, but we do not aspire to exaggerate. However, respecting the reviewer's suggestion we improve the text while revising it.

**Specific comments**

C l. 30-31: in some (slow) flow regimes, the impact of a landslide may be more a consequence of its total mass (size/volume) than a consequence of its velocity. And the velocity is not the prevailing parameter to estimate the impact force in this case. I think the statement at l. 30-31 should be qualified or, even better, this last sentence of the abstract can be removed (the impact force is not addressed in the current paper).

R: We understand the reviewer's concern. Our model and analytical solutions are general. Landslide velocity are relatively high. Most of the velocity solutions we have presented are on the order of 10s of m/s close to observed fast landslide motions, also explained in the text. The impact forces are proportional to the square of velocity. Here is where the velocity plays the crucial role in estimating the destructive power of the landslide. So, what we said is essential/fits well to the essence of the manuscript, and we would like to keep this sentence.

C l. 135: I don't understand $\alpha_s = 1$ ... I guess that we have: $\rho_{mixture} = \alpha_s \rho_s + (1 - \alpha_s)\rho_f$. The dry landslide case thus corresponds to $\gamma$ that tends toward 1 ($\gamma$ is equal to an epsilon when $\rho_f$ is the density of air and very small compared to the grain material density $\rho_s$ but is never zero in fact in an "air environment") and $\alpha_s$ is the volume fraction of the grains in air, which is restricted to the maximum random close packing (always smaller than 1) of the granular medium ? Please check/fix this.

R: Thanks for the comment. However, here we consider the mixture of solid and fluid (water plus colloids), but do not consider air. So, in the limit of vanishing fluid, the description of the limiting solid fraction as $\alpha_s = 1$ is consistent. And, since there is no buoyancy, $\gamma = 0$.

C l. 155: "genuinely" should be removed.

R: We agree to remove.

C: I'm not sure that section 2.3 is needed: there are many arguments presented here that are repeated all along the manuscript and in the section 6 (and 7) again. This section may be removed to shorten the manuscript/avoid repetitions. The main lines could be added after the key eq 5 (end of previous section 2.2) and that's it.

R: This section is very important and will have a lasting impact. It has been designed to clearly explain the physical, mathematical and engineering aspects of the main model (5) in a unified way. Furthermore, it describes how the new equation (5) extends the widely used classical non-linear inviscid Burgers' equation, classical shallow water equation, and the classical Voellmy model, beyond the current state. So, it is essential to keep this section. However, we still reduce its length as much as possible.

C l. 179: "... superior over..." I don't know if this a good statement to keep. This could be replaced by "... prior to...". I think the most important is to understand the foundation of the physical equations and make "them talk" by considering asymptotic solutions (first step) before launching sophisticated simulations based on those physical equations that are solved by complicated numerical schemes (second step). Sometime modelers forget the first step which is very useful to interpret complicated simulations and/or avoid mistakes in the second step. I would say for instance: "... and are often needed as a prerequisite before running numerical simulations based on complicated numerical schemes (yet based on the same physics in the end)."

R: We agree. Following the suggestion, we improve the text, which will read: "Physically meaningful exact solutions explain the true and entire nature of the problem associated with the model equation (Pudasaini, 2011; Faug, 2015), and thus, should be developed, analyzed and properly understood prior to numerical simulations. These exact analytical solutions provide important insights into the full flow behavior of the complex system (Pudasaini and Krautblatter, 2021), and are often needed to calibrate and validate the numerical solutions (Pudasaini, 2016) as a prerequisite before running numerical simulations based on complex numerical schemes. This is very useful to interpret complicated simulations and/or avoid mistakes associated with numerical simulations."

C l. 191-192: eq 6 reduces to the center of mass model for a dry landslide if $\gamma = 0$ (or $\rho_f$ is very small compared to $\rho_s$). Maybe this should be specified here.

R: The text will be enhanced as follows: "Classically, (6) is called the center of mass velocity of a dry avalanche of flow type (Perla et al., 1980)." changing to "Classically, (6) is called the center of mass velocity of a dry avalanche of flow type (Perla et al., 1980) for $\gamma = 0, \alpha_s = 1, K = 1$, and for negligible free-surface pressure gradient. This has been discussed in detail in Section 3.2."

C: figure 3: the 2 plots are already presented in figures 1 and 2. Either you remove figure 3 or I wonder whether you could consider instead some normalized versions of the plots: velocity/$Uo$ versus $x/Lo$ (top plot) and velocity/$Uo$ versus $t/To$ (bottom plot), where $Uo = \sqrt{(\alpha/\gamma)}$, $Lo = 1500$m, and $To = Lo/Uo$.

R: This figure will be removed.

C l. 273: "This is a fantastic situation." can be removed.

R: We think it is better to change it to "This is remarkable."

C l. 299: a strong shaking is an example of an initial input of strong kinetic energy but other situations are possible when a high potential energy is available and is converted quasi-instantaneously into kinetic energy (e.g. when the vertical drop of the detachment area is huge and combined with a high slope angle of the terrain).

R: Thanks for the suggestion. The text will be expanded incorporating your suggestion as: "e.g., by a strong seismic shacking, or when a high potential energy is available and is converted quasi-instantaneously into kinetic energy (the situation prevails when the vertical height drop of the detachment area is huge and the slope

angle of the terrain is high).”

C l. 334: “unique” can be removed or replaced by another word (like “interesting”) to tone down the statement.

R: We think it is better to leave this word here as it nicely fits to the text. We could not find other words to easily replace it with the intended meaning.

C l. 336 and 339: I’m note sure that “maximum” is adequate here. Why not using something like $U_s - s$ or $U_{infinity}$ for the steady-state value?

R: We understand the concern. The reason for choosing this terminology has been explained after (16) as “where, $u_{max}$ represents the maximum possible velocity during the motion as obtained from the (long-time) steady-state behaviour of the landslide” taking the upper limit of the velocity down the entire track. So, the term is meaningfully used.

C l. 344-354: this is a classical discussion, already addressed for the classical center of mass models. Please note that this discussion could be extended/updated by considering more recent approaches on center of mass models which considered some prescribed shapes for the profiles of the terrain, such as cycloidal and parabolic tracks (e.g. see Gauer CRST 2018).

R: Many thanks for providing the useful and relevant reference. We insert the following text while revising: “We mention that, for two-dimensional cycloidal or parabolic tracks, Gauer (2018) presented analytical velocities for the mass block motions with simple dry Coulomb or constant energy dissipation along the track. For such idealized path geometries he found an important relationship: that the maximum front-velocity, $U_{max}$, of major snow avalanches scales with the total drop height of the track, $H_{sc}$: $U_{max} \sim \sqrt{gH_{sc}/2}$, where $g$ is the gravity constant. Within its scope, this simple relationship may be applied to estimate the maximum velocity in (17).” [Gauer, P. (2018): Considerations on scaling behavior in avalanche flow along cycloidal and parabolic tracks. Cold Regions Science and Technology 151, 34-46.]

C l. 461-462: regardless of the model discussed, either the classical center of mass models or your model proposed here, I think that such models give upper bounds because they are not considering (by nature) the lateral spreading of the mass of the mixture when coming to rest. Such 3D (or 2D lateral spreading) effects when they are considered should give lower velocities and run-out. Please be careful and keep in mind that your more realistic model (I agree) can also give overestimates then. This is the reason why full 3D numerical simulations on a digital terrain model are needed too.

R: We agree. While revising, we improve the text by adding: “However, the reduced dimensional models and solutions considered here may give upper bounds to reality because they do not account for the lateral spreading of the landslide mass. Such problems can only be solved comprehensively by considering the numerical simulations on a full three-dimensional digital terrain model (Mergili et al., 2020; Shugar et al., 2021) by employing the full dynamical mass flow model equations (Pudasaini and Mergili, 2019) without constraining the lateral spreading.”

C l. 530: why choosing the powers 0.5 or 0.65? Is it fully arbitrary or do you have some arguments?

R: No, we can take any functions, as we have mentioned: “Any initial condition can be applied to the solution system (39)-(40)”. In Section 5.8, we have considered completely different function.

C l. 551-552: I agree that the initial conditions influence the dynamics over space/time but in the end the asymptotic states for sufficiently long distance (long times) are the same when the landslide come to a standstill, as shown in the two plots in figure 10. On other words, the way to go towards $U_0 = \sqrt{(\alpha/\gamma)}$ is not the same but $U_0$ is reached at the end of the day. Could you comment on this?

R: Somehow, it has been already explained. Yet, we improve the text by changing “This can be understood, because after a sufficiently long time, the motion is in steady-state. The two panels in Fig. 10 also clearly indicate that the stretching (rate) depends on the initial condition.” to “This can be understood, because

after a sufficiently long time, the motion is in steady-state. Nevertheless, the ways the two solutions reach the steady-state are different. The two panels in Fig. 10 also clearly indicate that the stretching (rate) depends on the initial condition."

C l. 585-586: "This is a fantastic situation.", again, can be removed!

R: This is so nice. So, we change "This is a fantastic situation," to "This is fascinating,", and hope that the reviewer will agree.

C l. 630: "This is a seminal understanding" is not needed I think. No need to oversell your work. The reader can appreciated its scientific quality by themselves.

R: We remove this sentence.

C l. 691-692: the last sentence of the paragraph should be removed.

R: We think this well summarizes the physical-mathematical fact and clearly/explicitly hints the readers for its originality. We ask the reviewer to allow us to keep this sentence.

C: I would suggest to remove the section 7 which repeats many statements already given either in the discussion section or along the main text of the manuscript. The manuscript is very long, difficult to digest. Section 7 is not needed.

R: Mentioned in R 3 above.

**Technical corrections/editing typos**

C l. 209: replace "has been" by "this will be".

R: Agree.

C l. 238-239: "... description. Both ..." Please revise, should be one sentence.

R: We put it in one sentence.

C l. 318: This section starts by "Second ... ", the first point being in previous sections (l. 312). This needs to be fixed. The two points should be in the same section. Please check and revise.

R: Thank you very much for this suggestion. We fix it by moving the last paragraph before Section 3.2.4 in to the first paragraph in Section 3.2.4.

C l. 387, l. 391 : note sure that "ms-2" is a correct notation; should be replaced by "m s-2" (empty space) or "m.s-2" (dot)?

R: We change it according to the recent publications in eSurf.

C l. 421-422: "So ... identify" can be removed; already said on l. 419 just above.

R: We remove it.

C l. 447: replace "Section 4.5" by "the current section" or "this section".

R: It should be "Section 4.6 and Section 4.7". So, we improve accordingly.

C l. 555: replace "has been" by "is".

R: We change it.

C l. 710: "large" repeated twice; please fix.

R: Will be fixed.

---

## Author Comment (AC3)

Dr. Jens Turowski
Associate Editor
Earth Surface Dynamics

**Revised Submission: MS # eSurf-2021-81**

Dear Dr. Turowski,
we very much appreciate the Reviewers and yourself for supporting our work. Our sincere thanks to the Reviewers for their time and constructive comments and explicit suggestions that resulted in the substantially improved manuscript in which we appropriately addressed all the concerns raised, including the presentation, length and the clarity of the paper. Included are the Response to Reviewer #1, Response to Reviewer #2, marked-up, and clear manuscripts. Please also see below the updated responses to reviewers which are now compatible with the revised manuscript. In the marked-up manuscript, the removed texts are in red and the edited/added texts are in blue color.

We hope that the revised manuscript will be suitable for publication in eSurf. We look forward to hearing from you soon.

With best regards,

Shiva P. Pudasaini, Michael Krautblatter

Technical University of Munich

**Response to Reviewer-1: MS #: eSurf-2021-81**

Reviewer's comments are denoted by C and our responses are denoted by R, respectively. In the marked-up manuscript, the removed texts are in red and the edited/added texts are in blue color.

**General comments**

C: This manuscript presented a simple and physics-based general analytical landslide velocity model which helps solve more landslide problems. The logic of the manuscript is clear, and the structure is reasonable, but there are still some problems.

R: We very much appreciate the Reviewer for supporting our work. Our sincere thanks to the Reviewer for your time and constructive comments and explicit suggestions that results in the substantially improved manuscript in which we appropriately address all the concerns you raised.

**Specific comments**

C1.: In my opinion, the title of this manuscript should be modified. You can make it more specific. It can't be seen from the current title that what you introduce in your article is related to landslide speed model.

R1.: Thank you very much for the suggestion. We can understand the reviewer's concern. We also thought to change the title to "A novel class of non-linear advective - dissipative system". We can be open to this choice. There are two major aspects of this manuscript. First, the development of the new Landslide Velocity equation (5), which, based on the physical parameters, involved forces and the dynamics, namely, the net driving and the resisting forces, presents a novel class of non-linear advective - dissipative system, the physical-mathematical model for landslide velocity. This has been exclusively discussed in Section 2.3. Second, construction of several novel exact analytical solutions to the model (5) for the velocity of the landslide. So, the overall essence of the manuscript is on the landslide velocity. The nice thing is that the same equation (5) can describe many different natural and physical phenomena by appropriately changing $u$ (the velocity) to any relevant state variable. Even the simplified version of (5); the equation (6) or (10); can describe wide range of physical phenomena, including: the Schrödinger equation, the Ermakov-Pinney, and the Friedmann equations in physical cosmology (https://arxiv.org/pdf/2112.11526.pdf). However, since our principle model is developed for velocity, and the manuscript is in the earth science journal, we think that the present title of the manuscript fits very well to what it describes. Also, the title as it stands now is nice. Whatever we call it, scientific communities may use the general exact analytical solutions constructed here to the context it fits to their interests.

C2.: Figure 3 is the combination of Figure 1 and Figure 2. There is no need to draw it again.

R2.: Thanks a lot for this legitimate suggestion. Fig. 3 will be removed, and the text improved accordingly.

C3.: The conclusion could have been a little more concise and organized. It can be divided into 1, 2 and 3 points.

R3.: We can make the Conclusion [Summary] much more concise and reduce it substantially by focusing only to the major outcomes. However, we think, it looks nicer in a plain text without dividing into points.

C4.: In Fig.10, Which graph represents the initial conditions $s_0(x) = x^{0.50}$ (top panel) or $s_0(x) = x^{0.65}$ (bottom panel)? In Fig.14, Which graph represents the initial conditions $\beta = 0.0019$ or $\beta = 0.019$? The different initial conditions should be represented in the diagram so that we can quickly distinguish between them.

R4.: The caption of Fig.10 will be improved, where, "The profiles correspond to the initial conditions $s_0(x) = x^{0.50}$ (top panel) and $s_0(x) = x^{0.65}$ (bottom panel), respectively." will read "The profiles evolve based on the initial conditions $s_0(x) = x^{0.50}$ (top panel, at $t = 0.0$ s) and $s_0(x) = x^{0.65}$ (bottom panel, at $t = 0.0$ s), respectively." In Fig.14, $\beta = 0.0019$ and $\beta = 0.019$ will be placed in the top and bottom figure panels, respectively. And, the figure caption will be improved consistently, from "The top panel with drag $\beta = 0.0019$,

while the bottom panel with higher drag, $\beta = 0.019$, which strongly controls the wave breaking and folding, and also the magnitude of the landslide velocity." to "The top panel with lower drag, while the bottom panel with higher drag, showing the drag strongly controls the wave breaking and folding, and also the magnitude of the landslide velocity."

C5.: In section 6.1: "the significant to large large geometric deformations" is not written correctly?

R5.: We change it to: "the significant to large geometric deformations"

**Response to Reviewer-2: MS #: eSurf-2021-81**

Reviewer's comments are denoted by C and our responses are denoted by R, respectively. In the marked-up manuscript, the removed texts are in red and the edited/added texts are in blue color.

**General comments**

C. The paper by Pudasaini & Krautblatter presents a series of detailed analytical solutions for the calculation of the velocity of a landslide, by considering the depth-averaged forms of incompressible mass and momentum conservation equations for a mixture of solid materials and a liquid (treated as a single-phase flowing material) as a starting point. It is shown that the analytic solutions though they are simplifications of the reality can capture many crucial mechanisms, such as the longitudinal stretching (the model remains 2D) of the mass during its movement along the slope, wave breaking and landslide folding. Moreover, simplified versions of the new analytic equations (equations numbered 11, 19 and 39-40 in the manuscript) allow to retrieve other simplified existing models, such as the ones given by solving the center of mass model developed by Voellmy –initially for snow avalanches– or the inviscid Burgers equations developed in fluid mechanics.

I found the scientific content of the paper very interesting with many interesting ideas discussed all along the development of these new analytic equations for the velocity of a landslide. Moreover I think that the paper is a niece piece of work for educational purposes, though I have some concerns regarding the presentation (see my general recommendation below).

The equations proposed by Pudasaini and Krautblatter remain a simplification of the reality but they take into account several key contributions to describe the motion of a landslide and thus are powerful tools to assess landslide risk and conduct relevant calculations for practitioners who are in charge of the hazard quantification and mitigation. They can complement more sophisticated calculations based on complicated numerical simulations, better than current analytic solutions based on center of mass models largely used in engineering (Voellmy-type models) can do.

As such, I have a very positive feeling on this work. However, I have a number of concerns that need to be addressed before I can definitely recommend it for publication.

The scientific content is very good and very interesting, overall. My concerns are on the presentation of the material. I must say that the paper was very difficult to digest (though I was excited to read it!). I really think the authors need to revise a lot the presentation to make the paper shorter and clearer.

R: We very much appreciate the Reviewer for supporting our work. Our sincere thanks to the Reviewer for your time and constructive comments and explicit suggestions that results in the substantially improved manuscript in which we appropriately address all the concerns you raised, including the presentation, length and the clarity of the paper. In the original submission, we thought we appropriately choose some words and phrases that would fit with the spirit of the text and would pleasure the readers, which however, will now be removed or made milder according to the reviewer's suggestions.

C: Here are some suggestions for that purpose:

C 1): the introduction is very very long: it really needs to be revised and shortened. There are a number of arguments repeated. I don't think that the authors need to oversell their results. They just need to present the facts and summarize them, this will be more efficient.

R 1): Thank you very much for this very reasonable suggestion. The Introduction was about one and a half page long in the initial submission, which we thought was ok for a professional paper. We do not intend to overvalue the work, but realize we need to considerably improve the presentation. However, we agree with the reviewer that there are number of statements repeated. Following your suggestion, we will present the facts and summarize them in a much more efficient and concise way resulting in substantially reduced Introduction.

C 2): I would suggest the authors to change the organization of the paper by starting with a presentation of the new analytic solutions: firstly with section 3.2 (eq 11), secondly with section 4 (eq. 19) and thirdly with section 5 (eqs 39-40). The simpler solutions for steady-state motion (eqs. 6, 7, 8) can be shortly presented as specific cases, after eq 11 or even in a supplementary material. The manuscript needs to be shortened.

R 2): We understand the reviewer's idea of first presenting the main general solutions, then collect all simpler solutions, or put them in a supplementary. We appreciate for this suggestion. However, the spirit of the ms is first to present simple solutions, similar to those available in literature, that many readers may be acquainted with. Then, tell what was not included or was not possible in the existing models, solutions and approaches, what we need to generalize and how we can achieve them. We presented the general solutions, then showed how those solutions can do more than existing ones, while in the mean time we analytically proved that our general solutions can directly recover all existing solutions as special cases. We think this way Sections 3-5 are well structured, smooth, easy to follow, also well formulated for educational uses. We hope the readers will like it. However, we have now identified many places where we can still substantially reduce Sections 3-5. We ask for the reviewer's support for this.

C 3): the summary section (section 7) is very long, with many repetitions again. And most of the material is already clear and said in the discussion section (section 6) or elsewhere in the paper. I think the summary section should be removed. If there are key ideas in the section 7 which the authors want to keep, they should be moved to the section 6 then and that's it.

R 3): We agree that there are many repetitions. In revising the ms, we remove all repetitions from Section 6, reducing it substantially. We will clearly re-write Section 6 to discuss the main results, while Section 7 will summarize the main findings, but without overlap. Moreover, we will largely reduce Section 7, making it much concise. This way it will be better.

C 4): there many superlatives or sentences that intend to praise the work done by the authors. Please let the readers themselves appreciate the quality of your work! I think those superlatives or statements that oversell your work are not needed and should be removed: see my suggestions in the list of specific comments below.

R 4): Some specific words were used to make it clearer the importance of the new findings as they are from the physical and mathematical ground, but we do not aspire to exaggerate. However, respecting the reviewer's suggestion we improve the text while revising it, and remove the suggested words and phrases.

**Specific comments**

C l. 30-31: in some (slow) flow regimes, the impact of a landslide may be more a consequence of its total mass (size/volume) than a consequence of its velocity. And the velocity is not the prevailing parameter to estimate the impact force in this case. I think the statement at l. 30-31 should be qualified or, even better, this last sentence of the abstract can be removed (the impact force is not addressed in the current paper).

R: We understand the reviewer's concern. Our model and analytical solutions are general. Landslide velocity are relatively high. Most of the velocity solutions we have presented are on the order of 10s of m/s close to observed fast landslide motions, also explained in the text. The impact forces are proportional to the square of velocity. Here is where the velocity plays the crucial role in estimating the destructive power of the landslide. So, what we said is essential/fits well to the essence of the manuscript, and we would like to keep this sentence.

C l. 135: I don't understand $\alpha_s = 1$ ... I guess that we have: $\rho_{mixture} = \alpha_s \rho_s + (1 - \alpha_s)\rho_f$. The dry landslide case thus corresponds to $\gamma$ that tends toward 1 ($\gamma$ is equal to an epsilon when $\rho_f$ is the density of air and very small compared to the grain material density $\rho_s$ but is never zero in fact in an "air environment") and $\alpha_s$ is the volume fraction of the grains in air, which is restricted to the maximum random close packing (always smaller than 1) of the granular medium ? Please check/fix this.

R: Thanks for the comment. However, here we consider the mixture of solid and fluid (water plus colloids), but do not consider air. So, in the limit of vanishing fluid, the description of the limiting solid fraction as $\alpha_s = 1$

is consistent. And, since there is no buoyancy, $\gamma = 0$.

C l. 155: "genuinely" should be removed.

R: We agree to remove. Line (L) 169 (of the revised marked-up ms).

C: I'm not sure that section 2.3 is needed: there are many arguments presented here that are repeated all along the manuscript and in the section 6 (and 7) again. This section may be removed to shorten the manuscript/avoid repetitions. The main lines could be added after the key eq 5 (end of previous section 2.2) and that's it.

R: This section is very important and will have a lasting impact. It has been designed to clearly explain the physical, mathematical and engineering aspects of the main model (5) in a unified way. Furthermore, it describes how the new equation (5) extends the widely used classical non-linear inviscid Burgers' equation, classical shallow water equation, and the classical Voellmy model, beyond the current state. So, it is essential to keep this section. However, we still reduce its length as much as possible (L169-181).

C l. 179: "... superior over..." I don't know if this a good statement to keep. This could be replaced by "... prior to...". I think the most important is to understand the foundation of the physical equations and make "them talk" by considering asymptotic solutions (first step) before launching sophisticated simulations based on those physical equations that are solved by complicated numerical schemes (second step). Sometime modelers forget the first step which is very useful to interpret complicated simulations and/or avoid mistakes in the second step. I would say for instance: "... and are often needed as a prerequisite before running numerical simulations based on complicated numerical schemes (yet based on the same physics in the end)."

R: We agree. Following the suggestion, we improve the text, which will read (L190-201): "Physically meaningful exact solutions explain the true and entire nature of the problem associated with the model equation (Pudasaini, 2011; Faug, 2015), and thus, should be developed, analyzed and properly understood prior to numerical simulations. These exact analytical solutions provide important insights into the full flow behavior of the complex system (Pudasaini and Krautblatter, 2021), and are often needed to calibrate and validate the numerical solutions (Pudasaini, 2016) as a prerequisite before running numerical simulations based on complex numerical schemes. This is very useful to interpret complicated simulations and/or avoid mistakes associated with numerical simulations."

C l. 191-192: eq 6 reduces to the center of mass model for a dry landslide if $\gamma = 0$ (or $\rho_f$ is very small compared to $\rho_s$). Maybe this should be specified here.

R: The text will be enhanced as follows (L213-214): "Classically, (6) is called the center of mass velocity of a dry avalanche of flow type (Perla et al., 1980)." changing to "Classically, (6) is called the center of mass velocity of a dry avalanche of flow type (Perla et al., 1980) for $\gamma = 0, \alpha_s = 1, K = 1$, and for negligible free-surface pressure gradient. This has been discussed in detail in Section 3.2."

C: figure 3: the 2 plots are already presented in figures 1 and 2. Either you remove figure 3 or I wonder whether you could consider instead some normalized versions of the plots: velocity/$Uo$ versus $x/Lo$ (top plot) and velocity/$Uo$ versus $t/To$ (bottom plot), where $Uo = \sqrt{(\alpha/\gamma)}$, $Lo = 1500$m, and $To = Lo/Uo$.

R: This figure will be removed, Page (P) 9.

C l. 273: "This is a fantastic situation." can be removed.

R: We think it is better to change it to "This is remarkable.", L297-298, otherwise we will remove it.

C l. 299: a strong shaking is an example of an initial input of strong kinetic energy but other situations are possible when a high potential energy is available and is converted quasi-instantaneously into kinetic energy (e.g. when the vertical drop of the detachment area is huge and combined with a high slope angle of the terrain).

R: Thanks for the suggestion. The text will be expanded incorporating your suggestion as (L324-328): "e.g., by

a strong seismic shacking, or when a high potential energy is available and is converted quasi-instantaneously into kinetic energy (the situation prevails when the vertical height drop of the detachment area is huge and the slope angle of the terrain is high)."

C l. 334: "unique" can be removed or replaced by another word (like "interesting") to tone down the statement.

R: We agree to remove, L365.

C l. 336 and 339: I'm note sure that "maximum" is adequate here. Why not using something like $U_s - s$ or $U_{infinity}$ for the steady-state value?

R: We understand the concern. The reason for choosing this terminology has been explained after (16) as (L367-368) "where, $u_{max}$ represents the maximum possible velocity during the motion as obtained from the (long-time) steady-state behaviour of the landslide" taking the upper limit of the velocity down the entire track. So, the term is meaningfully used.

C l. 344-354: this is a classical discussion, already addressed for the classical center of mass models. Please note that this discussion could be extended/updated by considering more recent approaches on center of mass models which considered some prescribed shapes for the profiles of the terrain, such as cycloidal and parabolic tracks (e.g. see Gauer CRST 2018).

R: Many thanks for providing the useful and relevant reference. We insert the following text while revising (L386-390): "We mention that, for two-dimensional cycloidal or parabolic tracks, Gauer (2018) presented analytical velocities for the mass block motions with simple dry Coulomb or constant energy dissipation along the track. For such idealized path geometries he found an important relationship: that the maximum front-velocity, $U_{max}$, of major snow avalanches scales with the total drop height of the track, $H_{sc}$: $U_{max} \sim \sqrt{gH_{sc}/2}$, where $g$ is the gravity constant. Within its scope, this simple relationship may be applied to estimate the maximum velocity in (17)." [Gauer, P. (2018): Considerations on scaling behavior in avalanche flow along cycloidal and parabolic tracks. Cold Regions Science and Technology 151, 34-46.]

C l. 461-462: regardless of the model discussed, either the classical center of mass models or your model proposed here, I think that such models give upper bounds because they are not considering (by nature) the lateral spreading of the mass of the mixture when coming to rest. Such 3D (or 2D lateral spreading) effects when they are considered should give lower velocities and run-out. Please be careful and keep in mind that your more realistic model (I agree) can also give overestimates then. This is the reason why full 3D numerical simulations on a digital terrain model are needed too.

R: We agree. While revising, we improve the text by adding (L500-505): "However, the reduced dimensional models and solutions considered here may give upper bounds to reality because they do not account for the lateral spreading of the landslide mass. Such problems can only be solved comprehensively by considering the numerical simulations on a full three-dimensional digital terrain model (Mergili et al., 2020; Shugar et al., 2021) by employing the full dynamical mass flow model equations (Pudasaini and Mergili, 2019) without constraining the lateral spreading."

C l. 530: why choosing the powers 0.5 or 0.65? Is it fully arbitrary or do you have some arguments?

R: No, we can take any functions, as we have mentioned (L572-573): "Any initial condition can be applied to the solution system (39)-(40)". In Section 5.8, we have considered completely different function.

C l. 551-552: I agree that the initial conditions influence the dynamics over space/time but in the end the asymptotic states for sufficiently long distance (long times) are the same when the landslide come to a standstill, as shown in the two plots in figure 10. On other words, the way to go towards $U_0 = \sqrt{(\alpha/\gamma)}$ is not the same but $U_0$ is reached at the end of the day. Could you comment on this?

R: Somehow, it has been already explained. Yet, we improve the text by changing (L606-610) "This can be understood, because after a sufficiently long time, the motion is in steady-state. The two panels in Fig. 10

also clearly indicate that the stretching (rate) depends on the initial condition." to "This can be understood, because after a sufficiently long time, the motion is in steady-state. Nevertheless, the ways the two solutions reach the steady-state are different. The two panels in Fig. 10 also clearly indicate that the stretching (rate) depends on the initial condition."

C l. 585-586: "This is a fantastic situation.", again, can be removed!

R: This is so nice. So, we change (L632) "This is a fantastic situation," to "This is fascinating,", and hope that the reviewer will agree, otherwise we will remove this phrase.

C l. 630: "This is a seminal understanding" is not needed I think. No need to oversell your work. The reader can appreciated its scientific quality by themselves.

R: We remove this sentence (L676).

C l. 691-692: the last sentence of the paragraph should be removed.

R: We remove this sentence (L737-738).

C: I would suggest to remove the section 7 which repeats many statements already given either in the discussion section or along the main text of the manuscript. The manuscript is very long, difficult to digest. Section 7 is not needed.

R: Mentioned in R 3 above.

**Technical corrections/editing typos**

C l. 209: replace "has been" by "this will be".

R: Agree (L232).

C l. 238-239: "... description. Both ..." Please revise, should be one sentence.

R: We put it in one sentence (L263-264).

C l. 318: This section starts by "Second ... ", the first point being in previous sections (l. 312). This needs to be fixed. The two points should be in the same section. Please check and revise.

R: Thank you very much for this suggestion. We fix it (L339-349) by moving the last paragraph before Section 3.2.4 in to the first paragraph in Section 3.2.4.

C l. 387, l. 391 : note sure that "ms-2" is a correct notation; should be replaced by "m s-2" (empty space) or "m.s-2" (dot)?

R: We change it according to the recent publications in eSurf.

C l. 421-422: "So ... identify" can be removed; already said on l. 419 just above.

R: We remove it (L459-460).

C l. 447: replace "Section 4.5" by "the current section" or "this section".

R: It was a mistake, it should be "Section 4.6 and Section 4.7". So, we improve accordingly (L485).

C l. 555: replace "has been" by "is".

R: We change it (L598).

C l. 710: "large" repeated twice; please fix.

R: Will be fixed (L756).

[revised manuscript text omitted]